

# OceanSODA-MDB: a standardised surface ocean carbonate system dataset for model-data intercomparisons

Peter E. Land[1], Helen S. Findlay[1], Jamie D. Shutler[2], Jean-Francois Piolle[3], Richard Sims[2], Hannah Green[2,1], Vassilis Kitidis[1], Alexander Polukhin[4], Irina I. Pipko[5]

[1]Plymouth Marine Laboratory, Prospect Place, West Hoe, Plymouth, PL1 3DH, UK
[2]University of Exeter, Centre for Geography and Environmental Science, Penryn, Cornwall. TR10 9FE
[3]IFREMER
[4]Shirshov Institute of Oceanology, 36, Nakhimovskiy prospect, Moscow, 117997, Russia
[5]V.I. Il'ichev Pacific Oceanological Institute FEB RAS, Vladivostok, 690041, Russia

*Correspondence to*: Peter Land (peland@pml.ac.uk)

**Abstract.** In recent years, large datasets of *in situ* marine carbonate system parameters (partial pressure of $CO_2$ ($pCO_2$), total alkalinity, dissolved inorganic carbon and pH) have been collated, quality controlled and made publicly available. These carbonate system datasets have highly variable data density in both space and time, especially in the case of $pCO_2$, which is routinely measured at high frequency using underway measuring systems. This variation in data density can create biases when the data are used, for example for algorithm assessment, favouring datasets or regions with high data density. A common way to overcome data density issues is to bin the data into cells of equal latitude and longitude extent. This leads to bins with spatial areas that are latitude and projection dependent (e. g. become smaller and more elongated as the poles are approached). Additionally, as bin boundaries are defined without reference to the spatial distribution of the data or to geographical features, data clusters may be divided sub-optimally (e. g. a bin covering a region with a strong gradient).

To overcome these problems and to provide a tool for matching surface *in situ* data with satellite, model and climatological data, which often have very different spatiotemporal scales both from the *in situ* data and from each other, a methodology has been created to group *in situ* data into 'regions of interest': spatiotemporal cylinders consisting of circles on the Earth's surface extending over a period of time. These regions of interest are optimally adjusted to contain as many *in situ* measurements as possible. All surface *in situ* measurements of the same parameter contained in a region of interest are collated, including estimated uncertainties and regional summary statistics. The same grouping is applied to each of the non-*in situ* datasets in turn, producing a dataset of coincident matchups that are consistent in space and time. About 35 million *in situ* data points were matched with data from five satellite sources and five model and re-analysis datasets to produce a global matchup dataset of carbonate system data, consisting of ~286,000 regions of interest spanning 54 years from 1957 to 2020. Each region of interest is 100 km in diameter and 10 days in duration. An example application, the reparameterisation of a global total alkalinity algorithm, is shown. This matchup dataset can be updated as and when *in situ* and other datasets are updated, and similar datasets at finer spatiotemporal scale can be constructed, for example to enable regional studies. The



matchup dataset provides users with a large multi-parameter carbonate system dataset containing data from different sources, in one consistent, collated and standardised format suitable for model-data intercomparisons and model evaluations. The OceanSODA-MDB data can be downloaded from https://doi.org/10.12770/0dc16d62-05f6-4bbe-9dc4-6d47825a5931 (Land

and Piollé, 2022).

## 1 Introduction

The ocean absorbs carbon dioxide ($CO_2$) from the atmosphere, which reacts with water to form a weak acid, carbonic acid. Through the marine carbonate system, carbonic acid then rapidly dissociates to form bicarbonate and hydrogen ions. The

marine carbonate system acts to buffer increases in hydrogen ions, in particular by combining with carbonate ions to form more bicarbonate ions. Over glacial timescales, weathering of carbonate rocks has maintained relatively stable ocean pH levels, but since the industrial revolution, the rate of uptake of anthropogenically released $CO_2$ has been too rapid for the natural system to keep pace, resulting in the phenomenon of Ocean Acidification (OA) (Doney et al., 2020). OA shifts the balance of marine chemistry such that there is increasing $CO_2$, decreasing pH, and decreasing carbonate ions. These shifts have been

shown to significantly alter many biological processes (Kroeker et al., 2013), with implications for food webs, ecosystem processes and ultimately ecosystem services on which humans rely (Gattuso et al., 2015; Doney et al., 2020).

Whilst there has been a rapid increase in the number of observations of the marine carbonate system over the past decade (e. g. SOCCOM, Rödenbeck et al., 2015; Williams et al., 2017), focusing especially on $CO_2$ uptake and OA, there remain large

gaps both in space and time, especially in more remote locations such as the Arctic (AMAP, 2018), where we also know there is significant variability and enhanced acidification in several parts of the Arctic (e. g. Polukhin, 2019). The longest *in situ* time-series stations for seawater $pCO_2$ and other OA-relevant parameters cover a temporal period of about 40 years (Bates et al., 2014), and around the globe there are only a handful of such time-series stations. Although more have since been established, these time-series stations highlight how different locations experience different drivers and differing levels of

variability (Bates et al., 2014). More recently, research communities have joined to form networks that increase data sharing, resulting in large collated datasets such as the Surface Ocean $CO_2$ Atlas (SOCAT, ~28 million surface observations, Lauvset et al., 2018; Bakker et al., 2016), the Global Ocean Data Analysis Project (GLODAP, ~79,000 surface observations, Lauvset et al., 2021), and most recently the Coastal Ocean Data Analysis Product in North America (CODAP-NA, Jiang et al., 2021).

Several efforts have been made to develop interpolation products that can be used to make global assessments of how the marine carbonate chemistry is changing in both space and time (e. g. Rödenbeck et al., 2015). These include neural network (Landschützer et al., 2016; Denvil-Sommer et al., 2019; Sasse et al., 2013), linear (e. g. Takahashi et al., 2014) and non-linear



regression (e. g. Watson et al., 2020) approaches. Similarly, model and interpolated observation-based data are routinely assessed against global *in situ* datasets (e. g. a requirement for inclusion within the observational ocean carbon data presented in Friedlingstein et al., 2020). Not only can *in situ* data and climatologies be used for assessing the marine carbonate system, but models, reanalyses and satellite Earth Observations (EO) are now frequently utilised either for their direct outputs or as inputs to algorithms. The ESA OceanSODA (Satellite Oceanographic Datasets for Acidification) project (https://esa-oceansoda.org) utilises a range of data sources, including EO, to input into empirical and machine learning algorithms, generating synoptic scale outputs of OA-relevant parameters. For example the OceanSODA-ETHZ product (Gregor and Gruber, 2021) reproduces the global surface carbonate system from 1985 to 2020. At present there is no one dataset that matches up these various data in time, treating all data in a consistent manner to minimise biases caused by differences in space and time sampling of each dataset. Here we present a new matchups dataset that addresses this need, with a particular focus on the surface (less than 10 m) carbonate system.

When attempting to collate large coincident datasets, data are often combined from diverse sources with different data densities, e. g. combining daily station data from a cruise with high frequency measurements from an underway system, or with daily satellite sea surface temperature (SST) at 1 km resolution. The differences in data density between the collated datasets can create biases when the data are used, favouring datasets or regions with high data density. This problem is often overcome by binning the data in a map projection, most simply into cells of equal latitude and longitude extent, but this can also cause biases because, in the example of equal latitude/longitude bins, the bins become smaller and more elongated as the poles are approached. The bin boundaries are also generally unrelated to the data, which may result in the data being divided sub-optimally, for example a bin boundary may pass through a large cluster of data, inappropriately dividing it.

To overcome these problems and to provide a tool for matching *in situ* data with other datasets such as satellite, model or climatological data, which often have very different spatiotemporal scales both from the *in situ* data and from each other, a methodology has been developed to group *in situ* data into 'regions of interest' (ROI), spatiotemporal cylinders consisting of circles on the Earth's surface extending over a period of time. These cylinders are positioned such that each contains as many *in situ* measurements as possible, with as little overlap between cylinders as possible. In this way, every *in situ* measurement is uniquely associated with one region of interest. All *in situ* measurements of the same parameter contained in a region are collated, including their estimated uncertainties, and regional summary statistics are calculated. After the ROIs have been defined using the *in situ* data, the other datasets are treated in the same way, collating measurements that lie within each *in situ* ROI, with their estimated uncertainties, and generating summary statistics. OceanSODA-MDB, a global matchups database (MDB), is presented here, consisting of ROI with a maximum diameter of 100 km and duration of 10 days. An example application is shown, reparameterising the global total alkalinity ($A_T$) algorithm of Takahashi et al., (2014) to give a new $A_T$ algorithm specific to the top 10 m. The MDB can be updated as and when *in situ* and other datasets are updated, and similar datasets at different spatiotemporal scales can be constructed, for example for regional studies.

Earth System
Science
Data

This document describes the datasets that are present within the MDB, including some brief information about data collection
and analysis, followed by a description of the methods used to create the MDB itself. We then provide some summary statistics
for the *in situ* and other data held within the MDB and a discussion of the benefits of this method. Finally we present the
reparameterization of the global $A_T$ algorithm of Takahashi et al., (2014) as an example application of the MDB.

## 2 Input datasets

*In situ* carbonate system variables included in the MDB (with which all other datasets are matched) are pH, total dissolved
inorganic carbon ($C_T$), total alkalinity ($A_T$) and partial pressure of $CO_2$ in water ($pCO_{2w}$, converted from fugacity and corrected
for temperature differences if necessary). At least one of these four key variables must be measured for a sampling event to be
included in the MDB. Measurement uncertainties and quality control (QC) flags, where available, are also included. The MDB
only includes surface measurements, defined as depth less than 10 m. Other *in situ* measurements which are included if they
are coincident with a measurement of one or more of the four primary variables (pH, $C_T$, $A_T$ and $pCO_{2w}$) include temperature
(T), salinity (S), sea-air difference in partial pressure of $CO_2$ and other *in situ* measurements such as nutrients and chlorophyll-
a concentration (Chl-a). In addition, the following values not directly associated with the sampling event are included: water
depth, distance from the nearest coast and monthly climatological temperature, salinity, dissolved oxygen, nitrate, phosphate
and silicate, all interpolated from global gridded climatologies to each data point. Where coincident $C_T$ and $A_T$ are available,
$pCO_{2w}$, pH and the saturation states of aragonite ($\Omega_A$) and calcite ($\Omega_C$) are calculated from $C_T$, $A_T$, T, S and depth. All
measurements of each of these variables in a ROI are collated and summary statistics generated. Other datasets not associated
with individual data points (satellite, model and reanalysis data, see Table 1) are collated for each ROI and their summary
statistics are added to the MDB.

Measurement uncertainties are either taken from the source literature or given default values based on GLODAP and SOCAT
default uncertainties. Default uncertainties for $C_T$ and $A_T$ are 4 µmol kg$^{-1}$ (Lauvset et al., 2021). For $pCO_{2w}$, the default
uncertainty is 5 µatm unless SOCAT data have been assigned a QC flag of A or B, in which case it is 2 µatm (Lauvset et al.,
2018). The default uncertainty for pH is 0.005 (Lauvset et al., 2021). The standard deviation of the measurements used to
calculate the climatological data is given as a proxy to uncertainty, and the assigned $C_T$ and $A_T$ measurement uncertainties are
propagated through the calculations to produce uncertainty estimates for the calculated $pCO_{2w}$, pH, $\Omega_A$ and $\Omega_C$.

The input data for the MDB primarily come from publicly available online datasets, with exceptions noted below. Details for
these input data are provided in Table 1. Below we briefly summarise the main methods for sample collection, analysis and
quality control for each of the datasets, with the exception of data input from the World Ocean Atlas (WOA18, Locarnini et
al., 2018; Zweng et al., 2019; Garcia et al., 2019a; Garcia et al., 2019b; Boyer et al., 2018), the Ocean Carbon and Acidification



Data System (OCADS, Jiang et al., 2021), the Global Surface pCO₂ Database (LDEO v2018, Takahashi et al., 2020), the
Global Ocean Data Analysis Project (GLODAPv2.2020, Olsen et al., 2020; Olsen et al., 2016; Key et al., 2015) and the Surface
Ocean CO₂ Atlas (SOCATv2020, Lauvset et al., 2018; Bakker et al., 2016) (SOCATv2020; Lauvset et al., 2018; Bakker et al.,
2016), as these have significant detail about data collection and quality control already described in the associated project
publications. Full details for all methods can be found in the associated references within this text and Table 1.

Biogeochemical-ARGO (Table 1, dataset no. 12) use profiling floats and measure pH using the Deep-Sea DuraFET, a sensor
comprising a Honeywell Ion Sensitive Field Effect Transistor (ISFET) and a chloride-ion-selective electrode as the reference
electrode, directly exposed to seawater (Claustre et al., 2020). We used delayed mode data where available, otherwise real
time mode, and used the estimated pH uncertainty given in the original data files.

Data from Plymouth Marine Laboratory (Table 1, datasets no. 6 and 7) were produced using an Apollo SciTech AS-C3 DIC
analyser for $C_T$, and using an Apollo SciTech AS-ALK2 analyser for $A_T$, following (Dickson et al., 2007). $C_T$ and $A_T$
measurements were calibrated using certified reference materials (CRMs) provided by A.G. Dickson from the Scripps Institute
of Oceanography. The precision and accuracy of replicate CRM analyses were better than ±2 µmol kg⁻¹. pH was determined
spectrophotometrically onboard the ship using m-cresol-purple dye (Clayton and Byrne, 1993) and again following best
practise (Dickson et al., 2007). The precision of triplicate pH samples was ±0.001 units or better.

Data from Woods Hole Oceanographic Institute (Table 1, dataset no 8) were produced using a Single-Operator Multimetabolic
Analyzer coulometer system for $C_T$ and using an open cell titration method with 0.1 N HCl for $A_T$ (Dickson et al., 2007), both
calibrated to Dickson CRMs (Scripps Institute of Oceanography). Pooled standard deviations for $C_T$ and $A_T$ were < 3.04 and
< 3.87 µmol kg⁻¹, respectively.

Data from Dalhousie University (Table 1, dataset no. 9) were produced using a Marianda Versatile Instrument for the
Determination of Titration Alkalinity (VINDTA) 3C coupled with a coulometer (UIC, Inc.) for $A_T$ and $C_T$ following standard
methods (Dickson et al., 2007). The instrument was calibrated against Dickson CRMs (Scripps Institute of Oceanography) and
the reproducibility of the $C_T$ and $A_T$ measurements was < 2 and < 3 µmol kg⁻¹, respectively.

Data from the Ocean Acidification Research Center at the University of Alaska Fairbanks (Table 1, datasets no 10 – 15) were
produced using a VINDTA 3C coupled with a coulometer (UIC, Inc.). Samples were standardized using Dickson CRMs
(Scripps Institute of Oceanography). Uncertainty for cruises ranged from 1 to 4 µmol kg⁻¹ for $A_T$ and 4 µmol kg⁻¹ for $C_T$. These
data are now included in the Coastal Ocean Data Analysis Product in North America (CODAP-NA), and hence have been
subjected to additional quality control (Jiang et al., 2021).



Data from the Shirshov Institute of Oceanology (Table 1, dataset no 16, not publicly available) were produced from samples collected in plastic 0.5 L bottles without preservation and analysed for pH and $A_T$. The pH value was determined on the ionomer "Ekoniks Expert 001" with a glass composite pH electrode by CJSC "Akvilon" (Moscow, Russia), calibrated using buffer solutions ISO 8.135-74 (techniques as per Dickson et al., 2007). Analysis of $A_T$ was conducted by direct titration (Bruevich, 1944) with a visual determination of the titration end point. This method, developed in the 1930s, shows very good correlation (Pavlova et al., 2008) with other methods of $A_T$ determination (Dickson et al., 2003; Edmond, 1970; Dickson and Goyet, 1994).

Data from the V. I. Il'ichev Pacific Oceanological Institute (Table 1, dataset no. 17, not publicly available) were produced using an indicator titration method in which 25 mL of seawater were titrated for $A_T$ with 0.02M HCl in an open cell (Bruevich, 1944; Pavlova et al., 2008), and a potentiometric method was applied to determine pH on the Pitzer pH scale (Pitzer and Press, 1991) using a closed cell held at constant 20 °C temperature with a sodium and hydrogen glass electrode pair without liquid junctions (Tishchenko et al., 2001; Tishchenko et al., 2011). $A_T$ measurements were performed with a precision of ~2 μmol kg$^{-1}$ with the accuracy set by calibration against Dickson CRMs (Scripps Institution of Oceanography). A TRIS–TRIS–HCl–NaCl–H2O buffer solution (Tishchenko et al., 2001; Tishchenko et al., 2011) was used for calibrations on the Pitzer pH scale. Both the hydrogen glass electrode and the sodium glass electrode were calibrated using this buffer. Together with thermodynamic data (Dickson, 1990), the pH values were converted from the Pitzer pH scale to the total hydrogen ion concentration scale ($pH_T$) (Tishchenko et al., 2001; Tishchenko et al., 2011; Dickson et al., 2007). The precision of pH measurements was about 0.004 pH units, with the accuracy set by calibration against buffer solution on the Pitzer pH scale.

Data from the Climate Change Initiative Sea Surface Temperature (CCI SST) Level 4 Analysis Climate Data Record (Table 1, dataset no. 18), produced by merging observations from satellite instruments NOAA Advanced Very High Resolution Radiometer (AVHRR) and ESA Along Track Scanning Radiometer (ATSR) using a data assimilation scheme, provide gap-free global daily fields of sea surface temperature (SST) at 0.2 m depth on a global 0.05° grid (Good et al., 2019; Merchant et al., 2019). We used the Version 2.1 dataset produced as part of the European Space Agency (ESA) Climate Change Initiative Sea Surface Temperature project from 1981 to 2016 and the complementary Version 2.0 dataset from the Copernicus Climate Change Service (C3S) from 2017 to 2020.

Data from the Climate Change Initiative Sea Surface Salinity (CCI SSS) Level 4 Analysis (Table 1, dataset no.19), produced by merging observations from satellite instruments ESA Soil Moisture and Ocean Salinity (SMOS) (January 2010–November 2019), NASA Aquarius (August 2011–June 2015) and NASA Soil Moisture Active Passive (SMAP) (April 2015–November 2019) using optimal interpolation, provide gap-free weekly maps of sea surface salinity (SSS) on a global 25 km EASE grid (Boutin et al., 2020; Boutin et al., 2021). We used the Version 2.31 dataset produced as part of the European Space Agency (ESA) Climate Change Initiative Sea Surface Salinity project from 2011 to 2019. Comparisons of the weekly Level 4 product



against Argo floats over the whole period and at global scale show a satellite – *in situ* bias of 0.0 and an RMSD of 0.28, while comparisons against thermosalinograph (TSG) measurements show a bias of -0.01 and an RMSD of 0.49. Under optimal conditions (Rain Rate=0 mm h$^{-1}$, 3 < 10 m wind speed < 12 m s$^{-1}$, SST > 5°C, > 800 km from coast), the bias and RMSD are

respectively 0.0 and 0.17 against Argo floats, and 0.0 and 0.18 against TSG (Boutin et al., 2021).

Data from the Arctic Sea Surface Salinity Level 3 composites (Table 1, dataset no 20), obtained from the Barcelona Expert Center (BEC, http://bec.icm.csic.es/), provide a daily weighted average of SMOS SSS in all overpasses over a 9-day period on a 25 km EASE grid centred on the north pole (Bec, 2021). We used the Version 3.1 dataset from 2011 to 2019. Comparisons

against Argo floats for the complete period show a bias of 0.02 and a RMSD of 0.39 (Olmedo et al., 2018).

Data from the SMAP Sea Surface Salinity Level 3 composites (Table 1, dataset no 21), produced by Remote Sensing Systems (RSS), provide a daily average of SMAP SSS in all overpasses over an 8-day period on a global 0.25° grid (Remote Sensing Systems, 2019; Meissner and Wentz, 2019). We used the Version 4.0 dataset from 2015 to 2021.


Data from the Climate Change Initiative Ocean Colour (CCI OC) Level 3 binned (Table 1, dataset no 22), produced by merging observations from satellite instruments NASA SeaWiFS (September 1997 to December 2010), ESA MERIS (April 2002 to April 2012), NASA MODIS (July 2002 to present), NOAA/NASA VIIRS (2012 to present), and ESA Sentinel 3A OLCI (May 2016 to present) using a blending method based on optical water type, provide Chl-a on a global 1/24° grid (Sathyendranath

et al., 2019; Sathyendranath et al., 2021). We used the Version 4.0 (1997-2019) and Version 5.0 (2019-2020) datasets produced as part of the ESA Climate Change Initiative Ocean Colour project. Comparisons against *in situ* measurements show a global mean bias in log$_{10}$(Chl-a) of −0.04 and RMSD of 0.34 (Sathyendranath et al., 2019).

Data from the NOAA Level 4 Analysis Climate Data Record (Table 1, dataset no. 23), produced by merging AVHRR satellite

data with measurements from ships, buoys and Argo floats using an optimal interpolation scheme, provide gap-free global daily SST on a global 0.25° grid (Huang et al., 2021; Banzon et al., 2016; Reynolds et al., 2007). We used the Version 2.0 dataset from September 1981 to December 2019, and Version 2.1 for 2020. As the analysis uses both night and day observations, it cannot be considered as foundation sea surface temperature and includes some diurnal warming effects.

Data from the Coriolis Ocean Dataset for Reanalysis (CORA) dataset (Table 1, dataset no. 24), produced from the merging of many different sources collected by Coriolis data centre in collaboration with the In Situ Thematic Centre of the Copernicus Marine Service (CMEMS INSTAC), acquired both by autonomous platforms (Argo profilers, fixed moorings , gliders , drifters, sea mammals) , research or opportunity vessels (CTDs, XBTs, ferrybox), provide monthly temperature and salinity on a global 0.5° grid (Szekely et al., 2019). CORA is a 4-dimensional dataset and we used only the temperature and salinity

from the first level (1 m depth). We used the Version 5.2 dataset from 1990 to 2019.



Data from the In Situ Analysis System (ISAS) dataset (Table 1, dataset no. 25), produced from the merging of the Argo network of profiling floats and other *in situ* sources using an optimal interpolation scheme, provide monthly temperature and salinity on a global 0.5° grid at several standard depth levels (Kolodziejczyk et al., 2021; Gaillard et al., 2016). We used only the

salinity from the first level (1 meter depth). We used the ISAS15 v7 dataset from 2002 to 2015 and ISAS20_ARGO v7 (Argo floats only) from 2016 to 2020.

Data from the OceanSODA-ETHZ dataset (Table 1, dataset no. 26), produced by ETH Zurich from surface ocean observations (SOCAT, GLODAP), using the newly developed Geospatial Random Cluster Ensemble Regression (GRaCER) method,

provide monthly $C_T$, $A_T$, $pCO_{2w}$ and pH on a global 1° grid (Gregor and Gruber, 2021). We used the $C_T$, $A_T$, $pCO_{2w}$, pH, temperature, salinity variables provided in the v2020b dataset from 1990 to 2018. For the open ocean, the estimated $pCO_{2w}$ and $A_T$ show global near-zero biases and root mean squared errors of 12 µatm and 13 µmol kg$^{-1}$, respectively. Taking into account also the measurement and representation errors, the total uncertainty increases to 14 µatm and 21 µmol kg$^{-1}$, respectively. Comparisons against direct observations from GLODAP show surface ocean pH and $C_T$ global biases of near

zero and root mean squared errors of 0.023 and 16 µmol kg$^{-1}$, respectively (Gregor and Gruber, 2021).

Data from the Copernicus Marine Service (CMEMS) Global Ocean Surface Carbon dataset (Table 1, dataset no. 27) is a reconstruction of monthly surface ocean $pCO_{2w}$, air-sea fluxes of $CO_2$ and pH with associated uncertainties on a global 1° grid. The product is obtained from an ensemble-based forward feed neural network, mapping SOCAT *in situ* surface ocean fugacity,

salinity, temperature, sea surface height, Chl-a, mixed layer depth and atmospheric $CO_2$ mole fraction. Surface ocean pH on the total scale is computed from $pCO_{2w}$ and reconstructed $A_T$ using the CO2sys speciation software. We used $pCO_{2w}$ and pH from the CMEMS MULTIOBS_GLO_BIO_CARBON_SURFACE_REP_015_008 dataset (Chau et al., 2020), from 1990 to 2019. Comparisons of $pCO_{2w}$ against SOCATv2021 show an absolute bias of 1.15 Pa and a RMSD of 1.86 Pa in the global open ocean. Comparisons of pH against data from GLODAPv2.2021 bottle data show an absolute bias of 0.017 and RMSD of

0.03 in the global open ocean.

## 3 Methodology

### 3.1 Pre-processing

Before grouping *in situ* data into ROIs, the different *in situ* datasets must first be merged into a single collated dataset and

sorted into date order. The largest *in situ* dataset (SOCAT) is first divided into yearly subsets. If the number of stations (unique sampling locations and times) in a year exceeds a threshold of $10^5$, it is subdivided into monthly subsets, and if a month exceeds $10^5$ stations, it is further subdivided into daily subsets. Each subset is then sorted, first by date and time, then by latitude and





finally by longitude. Each station is labelled with its data source and version (in this case SOCATv2020), its estimated uncertainty and QC flag, if available. For most datasets the latter is a World Ocean Circulation Experiment (WOCE) flag, but 265 in the case of SOCAT this is always 2, so the QC flag is a classification A (the best) to D (the worst included in the final SOCAT product), see Lauvset et al. (2018) for details. The next dataset (LDEO) is then similarly subdivided, sorted and then merged into the first to form a single dataset, continuing to subdivide where a yearly or monthly subset expands beyond $10^5$ data points. SOCAT and LDEO have many measurements in common, and in case of matching stations (defined by a separation < 1 km and < 30 s), the LDEO station is discarded. This completes the main global $pCO_{2w}$ datasets. The next dataset to be 270 merged is GLODAP, the main global $A_T$, $C_T$ and pH dataset, using only stations with sampling depth less than 10 m. Further variables from GLODAP that have been included in the MDB are pH at 25°C, dissolved oxygen, apparent oxygen utilization, nitrate, nitrite, silicate, phosphate, total and dissolved organic carbon and nitrogen and Chl-a. Again, the WOCE flag is always set to 2 in GLODAP, so in this case the QC flag is a classification indicating whether secondary QC has been performed on the data (1.0) or not (0.0). All other datasets use the WOCE flag where available, quoted as an integer, so the three types of 275 QC flag can be distinguished. All other *in situ* datasets are then merged in the same way, discarding $pCO_{2w}$ measurements if they match spatiotemporally with SOCAT or LDEO measurements already in the dataset, and $A_T$, $C_T$ or pH measurements if they match with GLODAP measurements.

Next, ancillary data are added to each station in the collated dataset. The distance from the nearest coast is spatially interpolated 280 from https://oceancolor.gsfc.nasa.gov/docs/distfromcoast/, and where not included in the original data, water depth is spatially interpolated from https://www.gebco.net/data_and_products/gridded_bathymetry_data/gebco_2019/gebco_2019_info.html. Climatological optimally interpolated T, S, nitrate, phosphate, silicate and dissolved oxygen and their standard deviations are all interpolated spatially and temporally from https://www.nodc.noaa.gov/OC5/woa18/woa18data.html. Wherever a station contains *in situ* $A_T$, $C_T$, T and S measurements, these are used to solve the carbonate system and provide estimates of $pCO_{2w}$, 285 pH, $\Omega_A$ and $\Omega_C$ with Monte Carlo uncertainty estimates. These are estimated as their mean and standard deviation from 100 runs of the SeaCarb package (version 3.2.12) with $A_T$ and $C_T$ values taken from a Gaussian distribution with mean and standard deviation equal to the measured value and its estimated uncertainty. In each run, the following SeaCarb options are also selected randomly from those with ranges of validity of temperature and salinity appropriate to the given data point, hence including the component of uncertainty arising from these choices:

k1k2 is selected from 'm10', 'm06', 'l' and 'r';

kf is selected from 'dg' and 'pf';

ks is selected from 'd' and 'k';

b is selected from 'l10' and 'u74'.

3.2 Creating the radial *in situ* data



The next step is to group stations into cylindrical spatiotemporal ROIs, each of which consists of a circle on the Earth's surface (all points with a great circle distance from the centre less than a given radius, assuming a spherical Earth) between two dates and times. For the global MDB, the ROIs are restricted to a maximum radius of 50 km (diameter 100 km) and a maximum temporal extent of 10 days. These limits are adjustable, e. g. smaller values might be more appropriate in a regional dataset

with high spatiotemporal variability. A ROI is the smallest spatiotemporal cylinder that can contain all of its associated *in situ* stations. The procedure is as follows:

    1.   Define the first ROI centred on and containing the first station. A ROI containing a single station is infinitesimally small. Add the new ROI to a 'regions' list.

    2.   Select the next station. If any ROIs in the regions list are more than 15 days older than the new station, they cannot

305           interact with a ROI to which the new station (or any subsequent station) is added, hence they can be stored and removed from the regions list.

    3.   Try to add the new station to each ROI in the regions list in turn, starting with the most recent.

           3a. If the new station is already within the ROI limits, add the station to the ROI and continue from step 2.

           3b. If the ROI can be expanded to contain the new station without exceeding the size limits, create a copy of

310               the ROI, expand it enough to add the station and add it to a list of 'found' ROIs for this station.

    4.   If the found list is not empty, check whether each found ROI overlaps with others in the regions list. If so, remove it from the found list.

    5.   If the found list is still not empty, add the new station to the found ROI that moved least relative to the limits, e. g. if a ROI moved temporally by 5 days or spatially by 50 km, the relative distance would be 0.5. Replace the original

315           ROI in the regions list with the expanded ROI and continue from step 2.

    6.   Add a new ROI centred on the new station to the regions list and continue from step 2.

When a ROI is stored, the following summary statistics of each value contained in the ROI are calculated: number of measurements; minimum; maximum; mean; median; sample standard deviation and interquartile range. This includes

uncertainty estimates, hence as well as the variability between measurements in a ROI, we also calculate statistics of the estimated measurement uncertainty associated with each measurement. The mean of pH variables is calculated geometrically, i. e. from the mean of $[H^+]$, but it should be noted that the standard deviation is that of pH, not of $[H^+]$. In addition to $pCO_{2w}$ treated normally, a further dataset is processed consisting of $pCO_{2w}$ corrected at each measurement to the mean SST of the ROI using

$pCO_{2w}$ at $SST_{ROI} = (pCO_{2w}$ at $SST) \exp[0.0433(SST_{ROI} - SST) - 4.35 \times 10^{-5}(SST_{ROI}^2 - SST^2)]$ (Takahashi et al., 2009). All data sources of each measurement type in the ROI are listed along with the number of measurements contributed by each source, e. g. a ROI might contain 10 $pCO_{2w}$ measurements from SOCAT and one from LDEO. ROIs are stored in NetCDF files using the trajectory format for ungridded data.



Sequential processing of all *in situ* data is a major task that would take several weeks to complete on a normal personal computer, a situation that is likely to worsen as data volume continues to increase. To speed up processing, we initiated ROI definition from different times. A start year and month is specified, and ROIs are defined using only *in situ* data starting from that time. In this way, the task can be divided into parallel processing streams. Initial ROI definitions in each stream are in general different from those that would be generated sequentially from the start, so care must be taken in combining ROIs

generated from different start times. The approach we have adopted is to allow processing from an earlier start time to continue past the start time of the next processing stream, creating two concurrent ROI sets covering the same time period. The concurrent period is inspected for sequences of ROIs that are identical in the two sets of results. If such a sequence is more than 10 days long, ROIs from the earlier stream before the identical sequence cannot overlap with those from the later stream after the identical sequence, and so it is safe to merge the streams. If an identical sequence more than 10 days long cannot be

found, ROIs from the earlier stream before the longest identical sequence can be compared individually with those from the later stream after the identical sequence to ensure that none overlap. The merged data are stored in yearly netCDF files.

### 3.3 Creating the OceanSODA-MDB matchup database

F*elyx* is a tool created to extract data from along-track, swath or gridded datasets such as Earth Observation (EO) data over

defined ROIs ([https://felyx.gitlab-pages.ifremer.fr/felyx_docs/](https://felyx.gitlab-pages.ifremer.fr/felyx_docs/)). *Felyx* is a free software solution, written in python, the aim of which is to provide Earth Observation data producers and users with an open-source, flexible and reusable tool to allow scientific analysis and performance monitoring of scientific data through subsetting over specific areas or matching up with *in situ* measurements. The development of *felyx* is supported by Copernicus, the European Union's Earth observation programme.

*Felyx* is used in this context to extract EO, model, climatology and re-analysis data within maximum-sized ROIs centred on the *in situ* ROIs. Given the time and location of a ROI, *felyx* is able, for each EO data source, to extract observation subsets within a 50 km radius and +/-5 days from the ROI's centre time and location (Fig. 18).

Hence matchup data are all extracted over the same size regions centred on matching *in situ* data. This methodology ensures that all data being compared (e. g. satellite and *in situ* observations) are treated as consistently and equally as possible, allowing

all uncertainties in all observations to be included within the analysis. The resultant radial matchup data (the output from the *felyx* system) are stored in the same NetCDF files used for the *in situ* data. For each averaged parameter, the mean, median, standard deviation, minimum, maximum, interquartile range and sample count of the observations found within the ROI's search area and time frame are calculated and provided. Finally, the output files are enriched with metadata for traceability, compliance to the Climate & Forecast Convention and self-description of the content, and the units are harmonized across the

different *in situ* and earth observation sources.



## 4 Data overview

The collated input dataset contains 34,912,843 individual stations, of which 34,839,413 (99.8%) contain $pCO_{2w}$, 24,474
(0.07%) contain $A_T$, 27,032 (0.08%) contain $C_T$ and 21,924 (0.06%) contain pH (note that stations may contain more than one
carbonate system parameter). Based on the ROI definition of 100 km radius and 10 days duration, this collated dataset resulted
in 285,822 ROIs, of which 272,753 (95.4%) contain $pCO_{2w}$, 13.595 (4.8%) contain $A_T$, 15,041 (5.3%) contain $C_T$ and 19,613
(6.9%) contain pH. Dates range from 1957 ($pCO_{2w}$) to December 2020 (pH), with steep increases in the number of
measurements in the 1990s and further increases in $pCO_{2w}$ measurements in the early 2000s and in pH measurements in the
2010s, the latter associated with the recent development of autonomous pH sensors such as Bio-ARGO (Fig. 1, note the
logarithmic scale). The recent reduction in $A_T$ and $C_T$ measurements may be associated with the situation that, unlike $pCO_{2w}$
and pH, all $A_T$ and $C_T$ measurements in the dataset are performed in the laboratory, resulting in a delay in data submission.
There may also be a reduction in support for core laboratory measurements as new autonomous measurements become
available, although some will still be necessary for validating and calibrating autonomous sensors.

The total number of measurements is not necessarily a good indication of representativity, especially with the advent of flow-
through $pCO_{2w}$ instruments which can collect many measurements spanning a small spatiotemporal range. In this respect, the
number of ROIs is a better guide. Fig. 2 (note the linear scale) shows the number of ROIs per year and the mean number of
measurements per ROI, which is typically around 2 except in the case of $pCO_{2w}$, for which it increases from around 10 in the
1980s to over 200 in 2019. Since the advent of Bio-ARGO, which typically delivers one surface pH measurement per ROI,
the mean number of pH measurements per ROI has decreased, and this is likely to continue as the number of Bio-ARGO floats
increases.

Figures 3 to 6 show the mean, standard deviation (where a ROI contains more than one measurement) and number of
measurements in each ROI over the whole dataset. Note that the standard deviation is only over a 10-day period, and so does
not show variability on longer timescales, such as seasonality or interannual variability. Note also that these plots include
~270,000 points in the case of $pCO_{2w}$ and more recent measurements overlay older ones, so specific features seen in these
plots should be checked in more detail. Variability is greatest in coastal regions and in parts of the Arctic, e. g. the Beaufort,
East Siberian and Laptev seas and between Greenland and Svalbard.

Figures 7 to 10 show the mean in each ROI divided into seasons. Strong $pCO_{2w}$ seasonality is evident in the northern
hemisphere, with high values in the northern Pacific and Atlantic in Jan-Mar and in the subtropics in Jul-Sep. Seasonality is
less clear in $C_T$ and pH, which may be due to the lower data density. This may also be due to the relatively smaller amplitude
of seasonal changes with respect to the mean, e. g. in (Kitidis et al., 2017) a 6% increase in $pCO_{2w}$ results in a 0.7% increase



in CT and 0.3% decrease in pH over 19 years. We would only expect strong seasonality in $A_T$ where there are large seasonal variations in salinity or strong terrestrial influence.

Figures 11 to 14 show the mean in each ROI divided into decades. As well as the increases in data density, the increase in $pCO_{2w}$ with time is clearly visible. The recent Bio-ARGO measurements can clearly be seen as a 'speckle' pattern in the

Southern Ocean in 2010-2020 pH.

**5 Example application**

To illustrate the use of the MDB , we present a reparameterization of the Takahashi et al., (2014) (T14) global algorithm for potential alkalinity (PA), which is equal to $A_T$ plus nitrate. For this we use *in situ* $A_T$ and SSS and WOA monthly climatological nitrate, since this is what would be needed to produce synoptic maps of $A_T$, e. g. from satellite or model SSS. The T14 algorithm

is clearly not intended for application to extreme coastal waters, of which there are many in the MDB, and algorithm uncertainties are expected to increase if we include waters with coastal or benthic influence, but T14 offer no criteria to distinguish these. Sasse et al., (2013) presented their own algorithms for $A_T$ and $C_T$, and they used the criteria that waters should be considered free from coastal influence ('marine') if greater than 300 km from the nearest shore with water depth greater than 500 m. These thresholds are global and likely to be over-conservative in many regions. Here we adapt these

criteria, aiming for a marine definition that is as inclusive as possible while not significantly compromising the fit found in highly marine waters, and providing a separate fit for the remaining coastal waters.

The T14 regions we use are shown in Fig. 15 and mapped on a 1° grid in Supplementary Data. We have eliminated overlaps in the original T14 regions, adjusted some boundaries to align more closely with geographical boundaries and added the

western and eastern basins of the Mediterranean Sea. In each region, we divided the data into marine and coastal domains, initially defining marine as having distance from the nearest shore greater than D = 300 km and water depth greater than Z = 500 m. The data in each domain were divided into up to 10 subsets for cross validation using the following methodology. If the number of marine data in a region is less than 10, the (up to) 10 points with greatest min(distance from coast / D, water depth / Z) are defined as marine, and D and Z decreased by the minimum required to ensure that the new marine data meet the

marine definition. The data are divided into years, with the time series from HOT and BATS being labelled as 'year' 0 and 1, respectively. If this results in more than 10 subsets, the subsets with lowest occupancy are combined, with the proviso that adjacent years cannot share the same subset. This prevents early years, which tend to have lower data density, from all being combined into a single subset. This continues until the number of subsets is reduced to 10. If the number of yearly subsets is less than 10, each data point is given its own subset unless this results in more than 10 subsets, in which case data are assigned

randomly to 10 subsets with as equal data numbers as possible. If the final number of subsets is less than five, fitting is not



attempted. If fitting occurs in one domain but not the other, the fit parameters are applied to the other domain (shown as brackets in Table 2).

Error analysis is done using cross validation, training the T14 relationship PA = A SSS + B on all but one of the subsets using
single value decomposition linear regression, then the error (difference between the resulting fit and the data) at each point of the remaining validation subset is recorded. This process is repeated using a different validation subset each time, giving an error for every data point. These errors are used to calculate the following summary statistics: root mean squared error (RMSE); mean absolute error (MAE); median absolute error (MDE); mean and median error (both measures of bias).

To check whether the marine definition was over-conservative, we first defined an acceptable level of degradation in marine RMSE ($RMSE_m$) in exchange for an expansion of the marine domain, $RMSE_{max} = max(RMSE_m + 0.1, RMSE_m * 1.01)$. We then repeated the following procedure iteratively. We sorted the coastal data by either distance from coast or water depth, then converted the points with maximum distance or depth to marine if their absolute error was not greater than $RMSE_m$, meaning that the addition of the points would not increase $RMSE_m$. D and Z were adjusted if necessary to ensure that the new marine
data obeyed the marine criteria, and marine and coastal fits and statistics were recalculated. If the resulting marine RMSE was less than the lowest $RMSE_m$ found so far, $RMSE_{max}$ was recalculated. Next, we tried going beyond the coastal data with absolute error greater than $RMSE_m$, continuing to lower distance or depth until the absolute error again exceeded $RMSE_m$, but with opposite sign. Adjusted fits were calculated separately for adjustments of distance and depth, and for both combined, and the new fit with lowest marine RMSE was accepted if the marine RMSE was no more than $RMSE_{max}$, again adjusting D and
Z and recalculating $RMSE_{max}$ if necessary. This procedure was repeated until it resulted in no change from coastal to marine.

Results are shown in Tables 2 and 3. Table 2 shows the region names, the RMSE of the original T14 and reparameterised algorithms in the marine and coastal domains, and the root mean squared difference (RMSdif) between the T14 and new algorithms, a measure of the extent to which results have changed due to the reparameterization. Note that the RMSE of the
original algorithm includes data used by T14 in the original fit and so may be an underestimate, while that of the new algorithm is calculated using cross validation. This can result in the new RMSE being greater than the T14 RMSE despite the refitting. Table 3 gives details of the algorithms, including the thresholds of distance from coast and water depth used to define the marine domain and the slope and intercept in each region and domain. Globally, the marine RMSE was reduced from 15 µmol kg$^{-1}$ (likely to be an underestimate, see above) using the original T14 algorithm to 12 µmol kg$^{-1}$, while in coastal waters the
RMSE was reduced from 32 to 23 µmol kg$^{-1}$ using only data for which T14 makes a prediction, but reduced further to 22 µmol kg$^{-1}$ when using all data.

The greatest marine RMSdif (49 µmol kg$^{-1}$) occurred in T14 region 3 (High Arctic), and the next greatest (20 and 18 µmol kg$^{-1}$) in regions 1 and 4 (West GIN Seas and Beaufort Sea), all other regions having marine RMSdif less than 11 µmol kg$^{-1}$. Note



that in some cases the coastal RMSdif is less than the marine RMSdif, suggesting that the T14 fit was dominated in these regions by data that we classify as coastal. The fits in regions 1 to 4 are shown in Fig. 16. In region 1 the tightly linear data used by T14 are not present in the MDB, and the few data that are consistent with the T14 relationship are all coastal, while the rest of the data (marine and coastal) follow similar relationships to those in region 2 (T14 Fig. 3 West and East GIN Sea). The data in region 3 are mostly marine, and while the data used by T14 mostly follow a relationship also seen in some of the

MDB data, most of the MDB data follow another relationship, barely seen in the T14 data and consistent with the relationships seen in region 2 (T14 Fig. 3 High Arctic and East GIN Sea). In region 4, the T14 and reparameterised relationships are quite consistent with each other, the large RMSdif being mainly due to the large variability in this region. T14 also note that some of their region 1 data follow the region 2 relationship, which they ascribe to eddies from region 2. One explanation might be that these eddies have become more frequent since the data used by T14, another that the water mass corresponding to region

2 (Atlantic waters flowing north into the Arctic) has expanded west into region 1 and north into region 3. To test this, Fig. 17 shows a map of regions 1, 2 and 3 with data points coloured red, green and blue in proportion to the probability density of a Gaussian distribution with mean and standard deviation equal to the T14 prediction and its RMSE in T14 regions 1, 2 and 3, respectively. Hence a red point would be consistent with the T14 region 1 prediction and inconsistent with those of regions 2 and 3, a white point would be consistent with all three predictions and a black point would be inconsistent with all three. This

map indicates that only a narrow strip close to the Greenland coast follows the T14 region 1 relationship, the rest being more consistent with region 2, that the region 2 relationship remains more plausible than the region 3 relationship up to about 86° N, and that the region 3 relationship performs poorly in the Beaufort Sea sector between 130 and 180° W.

This exercise has shown success in improving the fit of T14, incorporating new data, clarifying the distinction between marine

and coastal domains and providing an optimal fit in marine areas and a generally poorer fit in coastal areas, with estimated uncertainties.

**6 Discussion**

In order to stay relevant, the MDB must be regularly updated. Updates to the MDB can be made by adding new data to the existing ROIs, only creating new ROIs where necessary. This enables the MDB to be updated much more quickly and easily

that recalculating all ROIs from the beginning. As well as inclusion of new data, this process allows updating of data already in the MDB, e. g. if an existing dataset is reprocessed. The scale of the ROIs (100 km diameter, 10 days duration) is adjustable and regional MDBs may be created on smaller spatiotemporal scales in regions where this scale is inappropriate. This MDB is focused on the surface carbonate system and only ROIs containing carbonate system variables are included, but the same methodology can be used to create MDBs for other parameters of interest, such as methane or dimethyl sulphide or an MDB

based purely on a single parameter like salinity, or SST.

It should be borne in mind that oceanic processes can have strong effects on smaller scales than the MDB, for instance a growing phytoplankton bloom might significantly change surface $pCO_{2w}$ over a 10-day period. These effects are potentially detectable in the MDB through e. g. the standard deviation of $pCO_{2w}$, but in general will be masked by the averaging inherent

in the MDB method (the same is true for any gridding or averaging approach). Other effects such as surface temperature gradients affecting $pCO_{2w}$ (Woolf et al., 2016) will be similarly masked by averaging over the top 10 m within the current MDB. The MDB approach might be inappropriate if the subject of study is highly dependent on sub-ROI effects, such as a study of the depth variation of $pCO_{2w}$ in the surface layer, and in such cases it would be better to use individual measurements. It should also be borne in mind that averaging will not remove biases (e. g. regional or seasonal) from the *in situ* data, though

it will reduce stochastic noise.

The reparameterisation described in Section 5, though successful, could have been done with the original $A_T$ measurements from the original sources, probably with similar results, because the data density of $A_T$ measurements is typically low. Over 50% of MDB regions with $A_T$ contain only one $A_T$ measurement, and only 1% contain 10 or more. The real distinction between

the MDB approach and use of individual data points comes when comparing data with very high and low densities, which are more likely to be found in parameters that can be measured at high frequency, such as $pCO_{2w}$. To illustrate, consider the hypothetical example of a little-studied sea to which we wish to apply a simple model of constant $pCO_{2w}$. We have data from two cruises, one in winter making a transect of 10 discrete samples and the other in summer with an underway system making a transect of 1000 samples. The actual $pCO_{2w}$ has a mean of 400 µatm, but with a seasonal cycle from 375 to 425 µatm not

accounted for by our simple model. If the 10 winter measurements average to 375 µatm and the 1000 summer measurements to 425 µatm, simple averaging of all data gives an estimate of just under 425 µatm, while if the 1000 samples are binned into 10 regions of 100 measurements each, the correct average is found. Although this example is unrealistic, it is representative of the problems that can occur when fitting models to unevenly distributed data. The models are more complex and the deviations less obvious, but if there are systematic effects not captured by the model and the data density is greater towards one side of

the distribution of these effects, then the model becomes biased. The best way to overcome this is to identify the biasing effects and account for them (e. g. a seasonal split in the example), but some will always remain. The MDB approach lessens the effect of unaccounted biases by evening out differences in data density as consistently as possible.

## 7 Data availability

All data are freely available on the IFREMER data repository,

https://data-cersat.ifremer.fr/data/ocean-carbonate/oceansoda-mmdb/

DOI: 10.12770/0dc16d62-05f6-4bbe-9dc4-6d47825a5931 (Land and Piollé, 2022)

## 8 Code availability

Code will be made available by the authors on reasonable request.

## 9 Conclusions

Here we present a global dataset created using a novel methodology for combining different dataset types (e. g. *in situ*, model, satellite) onto the same spatial and temporal scales using 'regions of interest' (ROIs). This method gives advantages over previous methods, which predominantly grid data by latitude and longitude, as it provides a uniform spatiotemporal resolution across the globe and minimises biases created by differences in data density. We have collated a large global dataset comprised

primarily of publicly available *in situ*, satellite and climatological data, processed it using this methodology and presented summary statistics describing the results. The resulting matchup database (OceanSODA-MDB) is suitable for reparameterising empirical algorithms, as demonstrated here, but would also enable validation, evaluation and performance assessment of satellite data, re-analysis or model datasets, as well as model-data intercomparisons, by simply adding these data into the existing MDB.


### Author contributions

P. E. Land and J. D. Shutler developed the initial database concept and then P. E. Land created all methods for the *in situ* dataset collation and region definitions and then wrote the initial manuscript. H.Findlay collated all *in situ* datasets and identified the quality control procedures and uncertainty values. J-F Piolle developed all Felyx aspects and collated all satellite,

model and re-analysis data. H. Green and R. Sims tested the resulting database. V. Kitidis contributed AMT data. A. Polukhin has collected the Kara Sea dataset, screened the data and assessed the possibility of using Russian data due to differences in methods for determining pH and total alkalinity. I. Pipko collected Arctic data. J. Salisbury contributed $pCO_{2w}$ data. All authors contributed to the manuscript.

### Competing interest

None.

### Acknowledgements

This work was funded by the European Space Agency OceanSODA project, grant no. 4000112091/14/I-LG. The Surface Ocean $CO_2$ Atlas (SOCAT) is an international effort, endorsed by the International Ocean Carbon Coordination Project





(IOCCP), the Surface Ocean Lower Atmosphere Study (SOLAS) and the Integrated Marine Biosphere Research (IMBeR)
program, to deliver a uniformly quality-controlled surface ocean $CO_2$ database. The many researchers and funding agencies responsible for the collection of data and quality control are thanked for their contributions to SOCAT. We are also greatly indebted to all who contributed their efforts to the GLODAP project. We thank the UK National Environment Research Council for funding the Atlantic Meridional Transect (Climate Linked Atlantic Sector Science; CLASS). The work of A. Polukhin was supported by a grant for young scientists MK-3506.2022.1.5. Field data processing was partly supported by the Russian
Science Foundation (grant 21-17-00027 to I. Pipko).

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



| No. | Parameters | Type | Dataset source/ name (version) | Time period | Region (resolution) | References |
|---|---|---|---|---|---|---|
| 1 | pCO$_{2w}$, SST, SST | *In situ* data | SOCAT (2020) | 1957 - 2020 | Global | (Bakker et al., 2016) |
| 2 | pCO$_{2w}$, SST, SST | *In situ* data | LDEO (2019) | 1957-2019 | Global | (Takahashi et al., 2020) |
| 3 | A$_T$, C$_T$, pH, SSS, SST, N, P, Si, DO | *In situ* data | GLODAP (2.2020) | 1972 - 2019 | Global | (Olsen et al., 2020; Olsen et al., 2016) |
| 4 | pH, SSS, SST | *In situ* data | ARGO (downloaded 7th January 2021) | 2012-present | Global | (Claustre et al., 2020) |
| 5 | A$_T$, C$_T$, SSS, SST | *In situ* data | OCADS | 2003-2018 | Atlantic/Pacific | (Jiang et al., 2021) |
| 6 | A$_T$, C$_T$, pH, pCO$_{2w}$, SSS, SST, N, P, Si | *In situ* data | AMT | 1995-2019 | Atlantic | (Kitidis et al., 2017) |
| 7 | A$_T$, C$_T$, SSS, SST | *In situ* data | Arctic coastal data | 2012–2014 | Arctic | Findlay, *pers. comms.* |
| 8 | A$_T$, C$_T$, Chl-a, SSS, SST, DO, N | *In situ* data | Beaufort Gyre exploration project (Woods Hole Oceanographic Institution) | 2003- 2019 | Arctic | (Zhang et al., 2020) |
| 9 | A$_T$, C$_T$, SSS, SST | *In situ* data | Mackenzie Shelf | 2014 | Arctic | (Mol et al., 2018) |
| 10 | A$_T$, C$_T$, SSS, SST | *In situ* data | CHO_OC~1 | 2010-2014 | Arctic | (Wisdom, 2014) |
| 11 | A$_T$, C$_T$, SSS, SST | *In situ* data | EXPOCODE 33HQ20170826 | 2017 | Arctic | (Cross et al., 2020) |
| 12 | A$_T$, C$_T$, SSS, SST | *In situ* data | HLY1103 | 2011 | Arctic | (Mathis et al., 2016a) |
| 13 | A$_T$, C$_T$, SSS, SST | *In situ* data | 316n20090614 | 2009 | Arctic | (Cross et al., 2019) |
| 14 | A$_T$, C$_T$, SSS, SST | *In situ* data | 33HQ20080703 | 2008 | Arctic | (Mathis et al., 2016c) |
| 15 | A$_T$, C$_T$, SSS, SST | *In situ* data | 33HQ20080329 | 2008 | Arctic | (Mathis et al., 2016b) |
| 16 | A$_T$, C$_T$, SSS, SST | *In situ* data | Kara Sea dataset | 1993- 2004 | Arctic | (Wallhead et al., 2017) |
| 17 | A$_T$, C$_T$, SSS, SST | *In situ* data | Eurasian Arctic Ocean | 2006-2009 | Arctic | (Pipko et al., 2017) |
| 18 | SST | Satellite | cci_sst (2.1) | 1981-2020 | Global (0.05° daily) | (Good et al., 2019; Merchant et al., 2019) |



| 19 | SSS | Satellite | cci_sss (2.31) | 2010-2019 | Global (25km 7 day) | (Boutin et al., 2021; Boutin et al., 2020) |
|---|---|---|---|---|---|---|
| 20 | SSS | Satellite | arctic_sss (3.1) | 2011-2019 | Arctic (25km 9 day) | (Martínez et al., 2020a; Martínez et al., 2020b) |
| 21 | SSS | Satellite | remss_smap_sss (4.0) | 2015-2021 | Global (25km 8 day) | (Remote Sensing Systems, 2019; Meissner and Wentz, 2019) |
| 22 | Chl-a | Satellite | cci_oc_chloro-a (5.0) | 1997-2020 | Global (1/24° daily) | (Sathyendranath et al., 2019; Sathyendranath et al., 2021) |
| 23 | SST | Satellite and *in situ* re-analysis | noaa_sst (2.1) | 1981-2021 | Global (0.25° daily) | (Huang et al., 2021) |
| 24 | SST, SSS | *In situ* re-analysis | cora_temperature, cora_salinity (5.2) | 1950-2020 | Global (0.5° monthly) | (Szekely et al., 2019) |
| 25 | SST, SSS | *In situ* re-analysis | isas15_temperature, isas15_salinity (15) | 2002-2015 | Global (0.5° monthly) | (Kolodziejczyk et al., 2021; Gaillard et al., 2016) |
| 26 | $A_T$, $C_T$, $pCO_{2w}$, pH, SST, SSS | Calculated | ethz_ta, ethz_dic, ethz_pco2, ethz_ph, ethz_temperature, ethz_salinity (2020b) | 1985-2020 | Global (1° monthly) | (Gregor and Gruber, 2020; Gregor and Gruber, 2021) |
| 27 | $pCO_{2w}$, pH | Calculated | cmems_pco2, cmems_ph (015_008) | 1985-2019 | Global (1° monthly) | (Chau et al., 2020) |
| 28 | SST, SSS, DO, N, P, Si | Climatology | woa18_temperature, woa18_salinity, woa18_oxygen_o, woa18_nitrate_n, woa18_phosphate_p, woa18_silicate_i | 1955 - 2017 | Global (1° monthly) | (Locarnini et al., 2018; Zweng et al., 2019; Garcia et al., 2019a; Garcia et al., 2019b; Boyer et al., 2018) |

**Table 1: Input data sources for various parameters (SST = sea surface temperature, SSS = sea surface salinity,**
**Chl-a = chlorophyll a, DO = dissolved oxygen, N = nitrate, P = phosphate, Si = Silicate, $A_T$ = Total Alkalinity,**
**$C_T$ = total dissolved inorganic carbon). Lines 1 to 17 are *in situ* datasets which are used to create the regions of**
**interest, hence do not have unique dataset names. Lines 18 to 28 are generated using *felyx*, so each has a unique**
**dataset name.**






| Region | Quoted RMSE T14 | Marine | | | Coastal | | |
|---|---|---|---|---|---|---|---|
| | | RMSE T14 | RMSE new | RMSdif | RMSE T14 | RMSE new | RMSdif |
| 1 West GIN Seas | 6.1 | 21.0 | 6.5 | 20.1 | 46.6 | 23.6 | 40.9 |
| 2 East GIN Seas | 12.3 | 14.7 | 14.2 | 6.2 | 15.6 | 16.4 | 4.0 |
| 3 High Arctic | 16.8 | 56.5 | 30.1 | 48.6 | 40.6 | 32.9 | 31.9 |
| 4 Beaufort Sea | 60.5 | 34.4 | 29.9 | 18.4 | 69.0 | 57.0 | 42.0 |
| 5 Labrador Sea | 17.2 | 11.3 | 10.5 | 6.1 | 39.3 | 25.2 | 30.9 |
| 6 Sub-Arctic Atlantic | 6.7 | 6.9 | 6.4 | 2.9 | 53.3 | 29.2 | 47.5 |
| 7 N. Atlantic Drift | 6.5 | 7.9 | 7.8 | 2.2 | 48.5 | 43.5 | 25.4 |
| 8 Central Atlantic | 12.6 | 11.0 | 10.4 | 4.0 | 22.7 | 22.6 | 8.3 |
| 9 S. Transition Zone | 7.6 | 20.6 | 20.7 | 1.7 | 6.4 | - | 5.5 |
| 10 Sub-Polar Transition | - | - | 14.9 | - | - | 23.9 | - |
| 11 Antarctic (Atlantic) | 5.4 | 9.4 | 8.8 | 4.1 | 8.9 | 9.4 | 1.6 |
| 12 Kuroshio-Alaska Gyre | 9 | 13.8 | 11.8 | 7.6 | 33.9 | 33.6 | 9.9 |
| 13 N. Central Pacific | 14.7 | 14.8 | 11.9 | 9.1 | 34.6 | 18.4 | 30.0 |
| 14 Transition Zone | - | - | 14.0 | - | - | 11.8 | - |
| 15 Okhotsk Sea | 8.9 | 7.6 | 7.1 | 5.2 | 0.0 | 25.0 | 12.2 |
| 16 Transition Zone | - | - | 8.9 | - | - | - | - |
| 17 Central Tropical N. Pacific | 8.9 | 8.4 | 7.6 | 3.7 | 9.4 | 9.8 | 1.4 |
| 18 Tropical East N. Pacific | 9.7 | 11.7 | 5.9 | 10.4 | 21.4 | 16.1 | 15.5 |
| 19 Panama Basin | 8.6 | 44.2 | - | - | 21.4 | - | - |
| 20 Equatorial Pacific | - | - | 11.5 | - | - | 14.9 | - |
| 21 Central South Pacific | 9.4 | 12.1 | 12.5 | 1.5 | 7.7 | - | 3.8 |
| 22 E. Central South Pacific | 4 | 11.2 | 9.1 | 9.1 | 7.2 | 8.0 | 1.6 |
| 23 Sub-Polar S. Pacific | 7.8 | 8.3 | 8.2 | 3.2 | 4.1 | - | 2.1 |
| 24 Sub-Polar Transition | - | - | 12.2 | - | - | - | - |
| 25 Antarctic (Pacific) | 6.7 | 12.8 | 9.9 | 10.9 | 11.9 | 8.2 | 10.1 |
| 26 Main North Indian | 6.7 | 13.3 | 13.7 | 1.9 | 16.7 | 17.6 | 2.9 |
| 27 Red Sea | 6.3 | 6.1 | 8.7 | 2.6 | 16.7 | - | 2.9 |
| 28 Bengal Basin | 10.7 | 8.0 | 7.5 | 5.4 | 0.0 | 14.9 | 14.5 |
| 29 Main South Indian | 7.6 | 8.0 | 8.1 | 1.5 | 9.5 | 10.1 | 2.2 |
| 30 S. Indian Transition | 5.5 | 8.6 | 8.2 | 3.7 | 10.4 | 10.6 | 1.2 |
| 31 Sub-Polar Indian | - | - | 7.1 | - | - | 6.7 | - |
| 32 Antarctic (Indian) | 6.6 | 11.0 | 9.0 | 6.8 | 12.1 | 11.6 | 4.1 |
| 33 Circumpolar Southern Ocean | 9.1 | 8.2 | 3.7 | 7.9 | 12.1 | - | 4.1 |
| 34 Western Mediterranean | - | - | 14.6 | - | - | - | - |



| | | | | | | | |
|---|---|---|---|---|---|---|---|
| 35 Eastern Mediterranean | - | - | 8.3 | - | - | 22.8 | - |

**Table 2: Quoted and calculated RMSE values in each region defined by Takahashi *et al*. (2014) (T14) partitioned into coastal and marine subregions (see Table 3 for definitions). Calculated values are for the original T14 coefficients (RMSE T14), the coefficients recalculated using the MDB with cross validation (RMSE new) and the RMS difference between the two over the MDB data (RMSdif).**


| T14 | T14 | | Distance | Water | marine | | coastal | |
|---|---|---|---|---|---|---|---|---|
| region | Slope | Intercept | from coast | depth | Slope | Intercept | Slope | Intercept |
| 1 | 14.12 | 1796.2 | 300 | 500 | 54.13 | 418.9 | 45.61 | 716.2 |
| 2 | 59.57 | 232.0 | 251 | 207 | 58.92 | 248.5 | 55.04 | 390.1 |
| 3 | 27.30 | 1340.7 | 93 | 0 | 42.45 | 838.6 | 44.17 | 753.5 |
| 4 | 61.29 | 285.8 | 300 | 429 | 52.43 | 536.2 | 45.45 | 739.6 |
| 5 | 37.27 | 1016.2 | 290 | 0 | 0.74 | 2278.1 | 47.55 | 649.9 |
| 6 | 45.37 | 730.6 | 290 | 489 | 45.47 | 724.2 | 17.23 | 1710.1 |
| 7 | 45.30 | 733.0 | 163 | 0 | 44.73 | 751.0 | 30.20 | 1266.9 |
| 8 | 58.25 | 270.9 | 254 | 0 | 59.42 | 224.6 | 53.40 | 440.0 |
| 9 | 30.27 | 1259.4 | 0 | 0 | 28.58 | 1315.8 | - | - |
| 10 | - | - | 211 | 383 | 58.14 | 343.5 | 20.32 | 1622.9 |
| 11 | 57.78 | 367.8 | 248 | 500 | 59.36 | 318.4 | 60.16 | 288.5 |
| 12 | 44.88 | 724.8 | 298 | 0 | 40.14 | 891.2 | 48.16 | 627.0 |
| 13 | 79.92 | -395.7 | 263 | 0 | 60.61 | 246.4 | 54.48 | 444.9 |
| 14 | - | - | 272 | 0 | 29.71 | 1256.7 | 19.33 | 1609.5 |
| 15 | 59.37 | 301.4 | 0 | 500 | 76.37 | -257.8 | (76.37) | (-257.8) |
| 16 | - | - | 0 | 0 | 42.05 | 824.1 | - | - |
| 17 | 65.55 | 9.4 | 180 | 0 | 63.09 | 91.4 | 67.28 | -51.7 |
| 18 | 82.20 | -553.2 | 300 | 500 | 54.65 | 380.4 | 76.35 | -340.9 |
| 19 | 74.27 | -290.5 | 0 | 0 | - | - | - | - |
| 20 | - | - | 78 | 0 | 60.77 | 178.0 | 66.17 | -9.7 |
| 21 | 66.64 | -28.4 | 0 | 0 | 64.18 | 58.9 | - | - |
| 22 | 58.88 | 268.5 | 300 | 500 | 51.86 | 519.0 | 56.23 | 358.8 |
| 23 | 45.10 | 733.6 | 0 | 445 | 38.54 | 960.3 | - | - |
| 24 | - | - | 0 | 0 | -15.02 | 2812.9 | - | - |
| 25 | 81.69 | -450.8 | 294 | 293 | 52.89 | 525.6 | 68.13 | 13.4 |
| 26 | 57.07 | 302.3 | 224 | 0 | 56.73 | 312.4 | 58.06 | 269.8 |
| 27 | 26.27 | 1417.2 | 0 | 0 | 25.64 | 1443.0 | - | - |
| 28 | 39.66 | 894.2 | 300 | 0 | 28.24 | 1271.4 | (28.24) | (1271.4) |
| 29 | 65.03 | 20.0 | 107 | 184 | 65.54 | 3.6 | 63.22 | 81.5 |

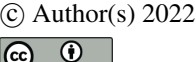



| 30 | 23.76 | 1486.1 | 300 | 0 | 28.40 | 1323.5 | 23.30 | 1500.7 |
| 31 | - | - | 193 | 0 | 5.54 | 2108.9 | 0.73 | 2279.8 |
| 32 | 62.57 | 202.0 | 87 | 0 | 37.37 | 1047.8 | 62.13 | 212.8 |
| 33 | 74.13 | -192.3 | 0 | 0 | 63.82 | 166.2 | - | - |
| 34 | - | - | 17 | 0 | 86.34 | -745.6 | - | - |
| 35 | - | - | 238 | 133 | 57.67 | 359.5 | 33.25 | 1317.6 |

**Table 3: Slope and intercept in each region defined by Takahashi *et al*. (2014) (T14). Original coefficients quoted by T14 are labelled T14, and reparametrized coefficients are given for coastal and marine subregions, where a point is marine if both distance from coast and water depth are greater than their regional thresholds.**

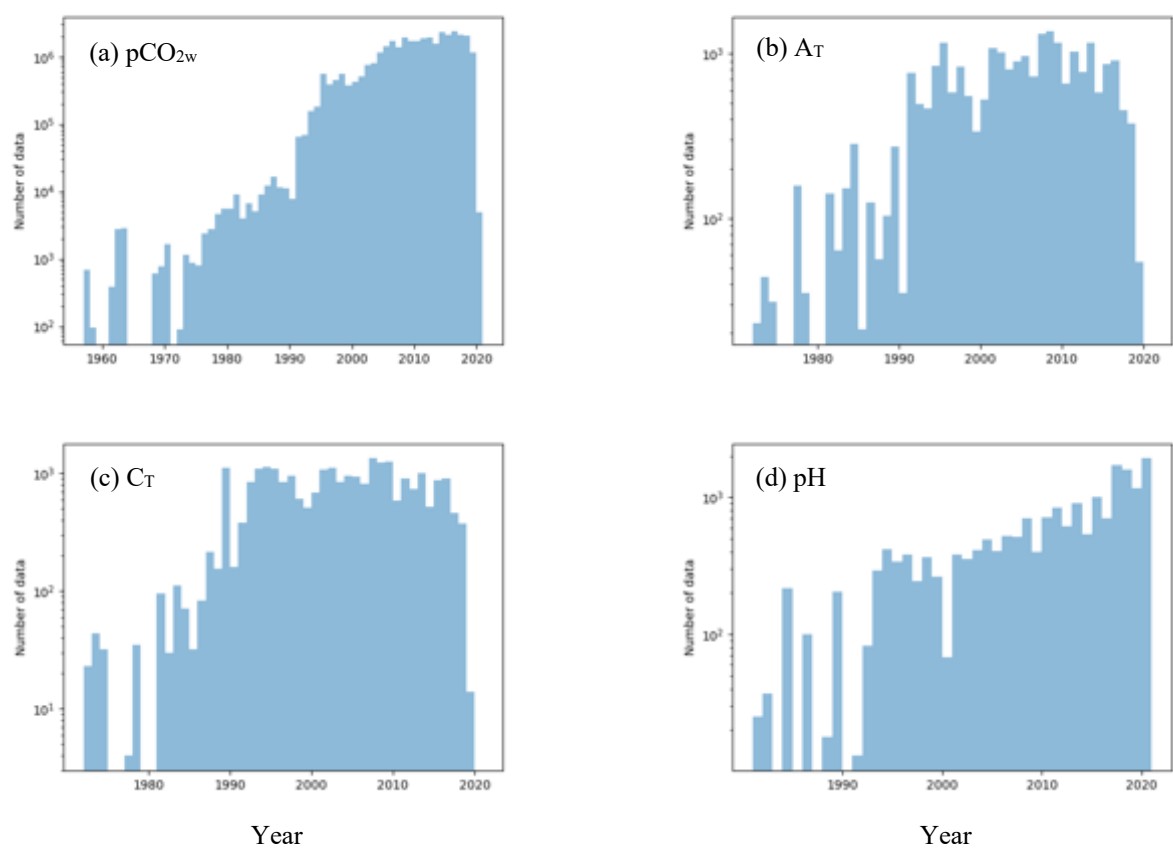


**Figure 1. Numbers of carbonate system measurements included in the database per year. (a) seawater pCO₂, (b) total alkalinity (A_T), (c) dissolved inorganic carbon (C_T), and (d) pH.**



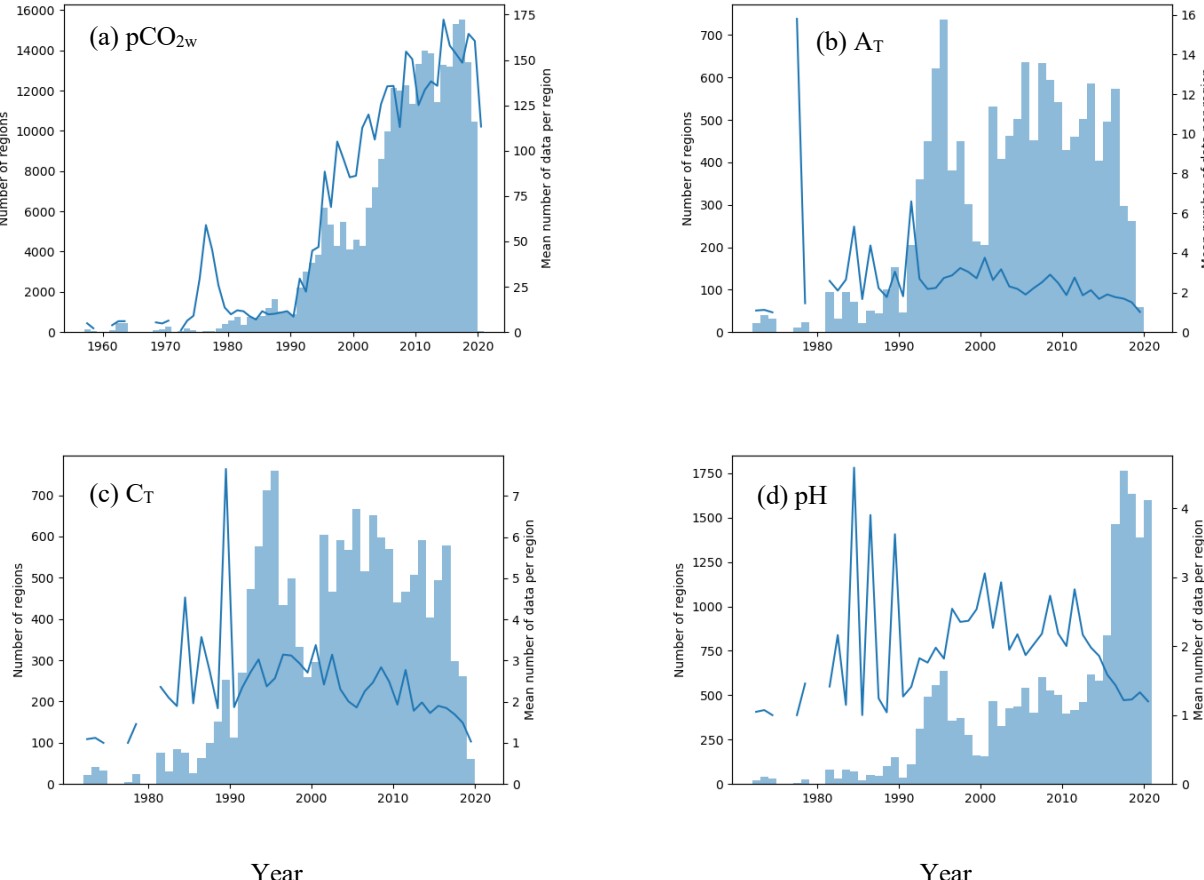

**Figure 2. Numbers of ROIs per year (bars) included in the database containing measurements of each carbonate system parameter, and the mean number of measurements per ROI (lines), for (a) pCO₂w, (b) total alkalinity (A_T), (c) dissolved inorganic carbon (C_T), and (d) pH.**





(a) pCO₂w ROI mean (μatm)

(b) pCO₂w ROI standard deviation (μatm)

(c) Number of pCO₂w measurements in each ROI

**Figure 3. Statistics of pCO₂w in each ROI over the whole database (272,753 ROIs). (a) ROI mean, (b) ROI Standard deviation, and (c) number of measurements in each ROI.**





(a) $A_T$ ROI mean (μmol kg$^{-1}$)

(b) $A_T$ ROI standard deviation (μmol kg$^{-1}$)

(c) Number of $A_T$ measurements in each ROI

**Figure 4. Statistics of $A_T$ in each ROI over the whole database (13,595 ROIs). (a) ROI mean, (b) ROI Standard deviation, and (c) number of measurements in each ROI.**






(a) $C_T$ ROI mean (μmol kg$^{-1}$)

(b) $C_T$ ROI standard deviation (μmol kg$^{-1}$)

(c) Number of $C_T$ measurements in each ROI

**Figure 5. Statistics of $C_T$ in each ROI over the whole database (15,041 ROIs). (a) ROI mean, (b) ROI Standard deviation, and (c) number of measurements in each ROI.**



(a) pH ROI mean (total scale, p=0, T=25°C)

(b) pH ROI standard deviation (total scale, p=0, T=25°C)

(c) Number of pH measurements in each ROI

**Figure 6. Statistics of pH in each ROI over the whole database (19,613 ROIs). (a) ROI mean, (b) ROI Standard deviation, and (c)**
**number of measurements in each ROI.**



Figure 7. Mean pCO$_{2w}$ (µatm) divided into seasons. (a) January – March (70,658 ROIs); (b) April – June (67,631 ROIs); (c) July – September (69,083 ROIs); (d) October – December (65,381 ROIs).

(a) Jan-Mar

(b) Apr-Jun

A$_T$ ROI mean (µmol kg$^{-1}$)

(c) Jul-Sep

(d) Oct-Dec

**Figure 8. Mean A$_T$ (µmol kg$^{-1}$) divided into seasons. (a) January – March (3602 ROIs); (b) April – June (3682 ROIs); (c) July – September (3960 ROIs); (d) October – December (2351 ROIs).**

Figure 9. Mean $C_T$ (µmol kg⁻¹) divided into seasons. (a) January – March (4029 ROIs); (b) April – June (4256 ROIs); (c) July – September (4197 ROIs); (d) October – December (2559 ROIs).




(a) Jan-Mar    (b) Apr-Jun

pH ROI mean (total scale, p=0, T=25°C)

(c) Jul-Sep    (d) Oct-Dec

**Figure 10. Mean pH (total scale, p=0, 25°C) divided into seasons. (a) January – March (5501 ROIs); (b) April – June (5386 ROIs); (c) July – September (4959 ROIs); (d) October – December (3767 ROIs).**
**Figure 11. Mean pCO$_{2w}$ (µatm) divided into decades: 1751 ROIs to 1969; 1636 in the 1970s; 9090 in the 1980s; 42,548 in the 1990s; 97,313 in the 200s; 120,415 from 2010 to 2020.**



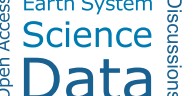

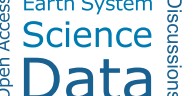

**Figure 12. Mean A_T (μmol kg⁻¹) divided into decades: 128 ROIs in the 1970s; 707 in the 1980s; 3923 in the 1990s; 5196 in the 2000s; 3641 from 2010 to 2020.**


**Figure 13. Mean C$_T$ (µmol kg$^{-1}$) divided into decades: 123 ROIs in the 1970s; 971 in the 1980s; 4613 in the 1990s; 5672 in the 2000s; 3662 from 2010 to 2020.**



**Figure 14. Mean pH (total scale, p=0, 25°C) divided into decades: 120 ROIs in the 1970s; 672 in the 1980s; 3428 in the 1990s; 4632 in the 2000s; 10,761 from 2010 to 2020..**

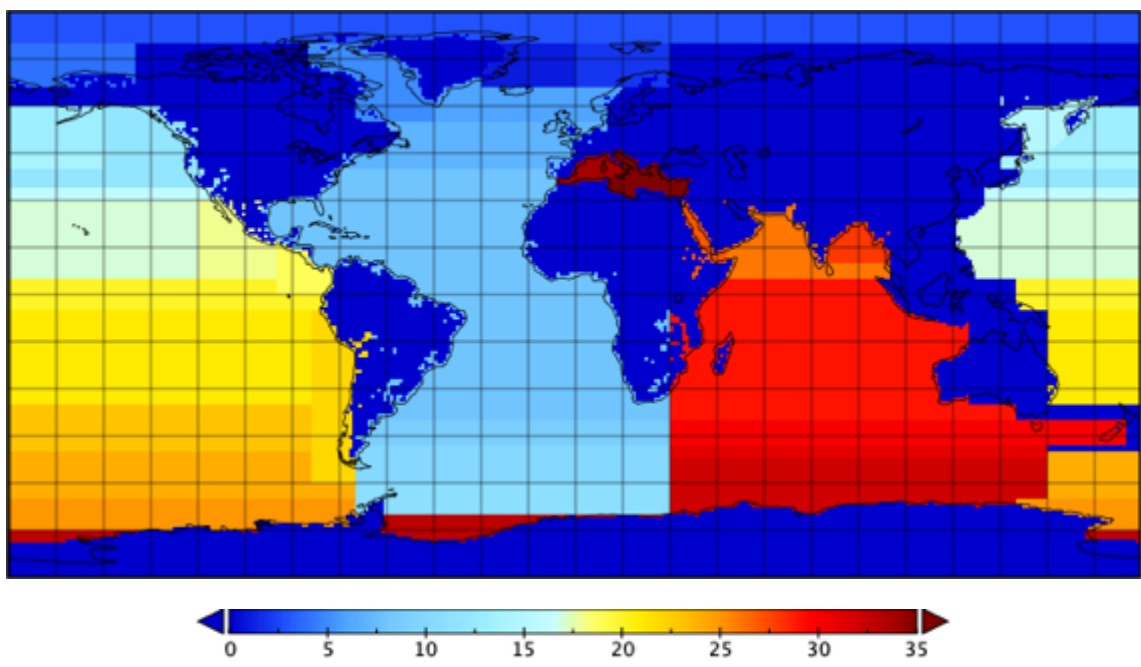


**Figure 15. Regions used in reparameterization of Takahashi *et al* (2014). See Table 2 for region names.**


**Figure 16. PA-SSS relationships in the first four T14 regions, all in the Arctic. Orange data are classified as marine, based on the distance from the nearest coast in km and the depth in m both being greater than region-specific thresholds. All other data (in blue) are classified as coastal. The thick green line is the T14 relationship and the thin green lines show the T14 quoted RMSE. The red and purple lines are the new fits to marine and coastal data. (a) Region 1, West GIN Sea; (b) region 2, East GIN Sea; (c) region 3, High Arctic; (d) region 4, Beaufort Sea.**


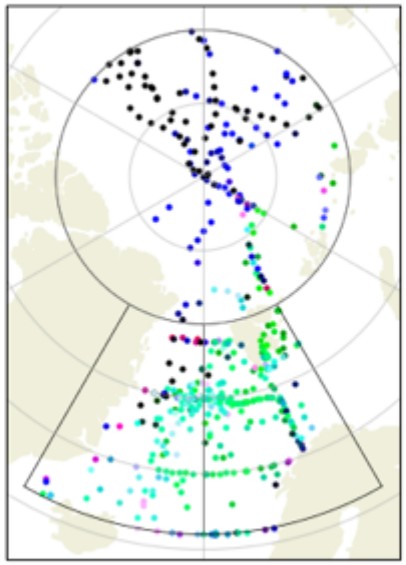

**Figure 17. T14 regions 1 (West GIN Sea, bottom left), 2 (East GIN Sea, bottom right) and 3 (High Arctic, top) with corresponding MDB data. Each point is coloured according to the probability density of a Gaussian with mean equal to the T14 regression and standard deviation equal to its quoted RMSE, which is a measure of how consistent the data are with the T14 fit. The red component is the consistency with the region 1 fit, green with the region 2 fit and blue with the region 3 fit.**




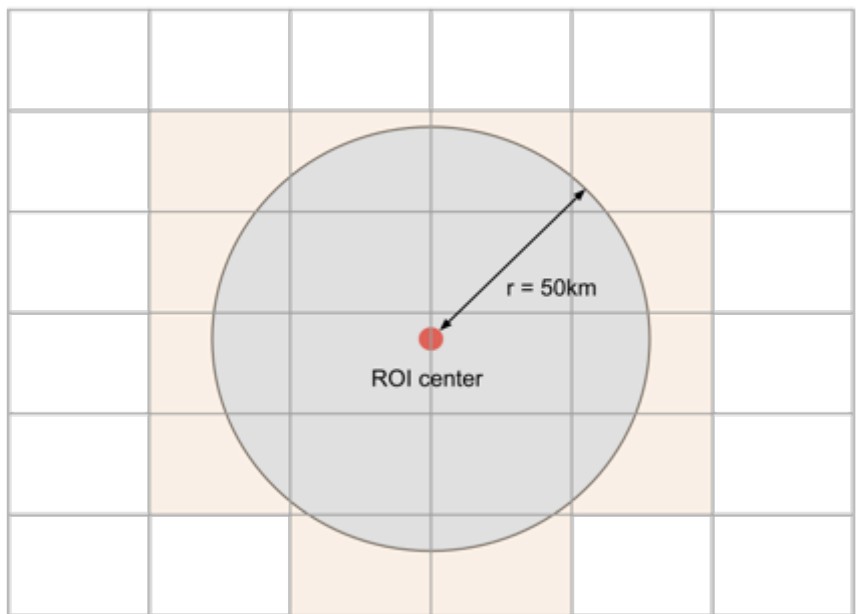

**Figure 18: spatial colocation principle for Earth Observation data. All gridded observations within or intersecting a**
**50 km radius from the ROI centre for all the consecutive files within +/- 5 days around the ROI centre time are**
**averaged together. The mean and other statistics (median, standard deviation, minimum, maximum and interquartile**
**range are also calculated and provided in the output matchup dataset.**