# Peer review of "OceanSODA-MDB: a standardised surface ocean carbonate system dataset for model-data intercomparisons"

_Earth System Science Data, 2022_

## Author Comment (AC1)

We thank the reviewers for their helpful and constructive comments. Author replies in red.

**RC1: 'Comment on essd-2022-129', Anonymous Referee #1, 20 Sep 2022**

The authors present a matchup database combining various in situ observations with associated satellite remote sensing and reanalysis products. The challenge of combining these datasets which are collected over vastly different spatiotemporal scales is handled by the novel development of spatiotemporal regions of varying size.

This technique can clearly be applied to other parameters, and the dataset has the potential for reuse.

The dataset is a novel combination of other datasets, some of which have never previously been made publicly available.

Data is accessible as described and easily downloaded without the need of registration. A subset of these data was downloaded for inspection. These netcdf-v4 compatible files were inspected using the ncdf4 R library.

the "_FillValue" for the variables consists of 8-bytes which is not a valid type for R and these are coerced to a double precision value, but otherwise no issues. The files are well described in typical netcdf format and adhere to CF conventions.

The files were also tested in a python 3.8 environment using the netCDF4 package.

The dataset is missing earlier years, 1971, 1964-1967 and 1959. I presume this is due to lack of associated in-situ data, but please confirm.

We can confirm that the missing years are due to the absence of in situ data in those years.

No erroneous data were found and the carbonate system and associated parameters (e.g. nutrients, temperature) are shown to span appropriate ranges and be consists. At least to the extent of my knowledge of the global distribution of these parameters.

The abstract is a little long. It includes reference to an example application which is not appropriate for an abstract and is also "not shown" - line 30. This appears to have been partially copied from the Introduction section. The article appears otherwise well written.

The example application is described in detail in Section 5 and is included simply to illustrate the use of the MDB. It is mentioned in the abstract because it is an entire section of the paper. We will change 'show' to 'present' in the abstract.

Each sub-source of data is well described, with collection method, calibration information and uncertainties. However, In the contributions is it mentioned that A. Polukhin assessed the possibility of using certain data due to differences in methods. These differences are not

well discussed in the input datasets section so I suggest the inclusion of a short paragraph summarising the rationale for this work.

We will include the following paragraphs:

In 2000, the WG13 (founded in 1998 under the PICES North Pacific Marine Science Organization) carried out an experiment on the intercalibration of the total alkalinity measurements in seawater with the first participation of Russian specialists, who presented the procedure by Bruevich. Within the experiment, twelve laboratories were involved: six from Japan, three from the USA, and one from Canada, South Korea, and Russia. For the intercalibration, the following methods were presented: potentiometric titration in a closed  and open cell, the single addition procedure by Culberson, and the method by Bruevich. The alkalinity examined in the samples was determined to ±1 µM/kg at the laboratory of Prof. Dickson. The intercalibration results showed that the Bruevich method agrees well with the commonly used procedures of potentiometric titration in a closed or open cell as well as with the single addition method.

Thus, the results of the intercalibration showed that the alkalinity values obtained by the Bruevich method using modern burettes, $Na_2CO_3$ of high purification degree as a standard to establish the acid titre, and applying correction for the acid density and for the weight of the acid and seawater samples in vacuum are in agreement with the standard within ±1 µM/kg. Under field conditions, the usual accuracy of the method for seawater analyses is equal to ±2.5 µM/kg. The method presented is easy and well applicable to the microanalysis of interstitial waters of marine sediments (Pavlova et al., 2008).

(full link Pavlova, G.Y., Tishchenko, P.Y., Volkova, T.I.,  Dickson A. & Wallmann K. Intercalibration of Bruevich's method to determine the total alkalinity in seawater. *Oceanology* 48, 438–443 (2008). https://doi.org/10.1134/S0001437008030168)

We note the AMT and Kara Sea datasets start over a decade before the publication of the Dickson et al 2007 best practice document which is given as a reference. I presume the authors are simply saying that the methods are basically the same, but it would be good to clarify.

Yes, we used essentially the same methods before Dickson et al wrote a guide. We will clarify this in the text. Both datasets have been published in peer-reviewed publications since Dickson et al 2007.

Line 486 - when and by whom will reassess the appropriateness of the maximum ROI scales? community feedback?

The spatiotemporal resolution of the ROIs was initially chosen as a good starting point based upon our knowledge of the datasets used, the likely space and time scales of

changes in these data and conditions, and the desire to ensure we have matchups between the insitu ROIs and the other datasets. Based on user feedback at a stakeholder workshop we are developing the next version of the MDB, which will be provided in the future as an updated version of this manuscript and data.

The figures provide a good overview of the nature of the dataset and are of generally good quality.

Figure 1 and 2 could be arranged better, the axis labels very small considering the amount of whitespace between panels

We will amend Figs 1 and 2 as suggested.

**RC2: 'Comment on essd-2022-129', Anonymous Referee #2, 27 Sep 2022**

General Comments

Over the past couple decades, large datasets of in situ carbonate system measurements (e.g., SOCAT, Bakker et al., 2016) have been compiled and quality controlled, and are frequently used to train models for prediction and/or spatiotemporal gap filling of carbonate chemistry using proxy variables. However, models trained on sparse and irregularly spaced data can suffer from biases that favor more highly sampled regions or time periods. These biases can be partially alleviated by aggregating data points to bins of constant spatial and temporal extent, but binning in this way can still result in sub-optimal data division due to elongation of bins near the poles, unintentional splitting of clustered data, and unaccounted for interactions with coastlines.

To address these shortcomings, Land and coauthors compile and describe a database of in situ surface carbonate chemistry measurements built around regions of interest (ROIs), which are constructed so as to include as many in situ measurements as possible within a maximum timespan of 10 days and a maximum diameter of 100 km. This database — OceanSODA-MDB — also includes spatiotemporal matchups with satellite, model, and reanalysis datasets corresponding to each ROI. Land et al. display the utility of this newly compiled database by re-training a global algorithm from Takahashi et al. (2014) to predict potential alkalinity (PA) from sea surface salinity, displaying a global reduction in the root mean squared error between measured and predicted PA: from 15 to 12 µmol/kg in marine waters and 32 to 23 µmol/kg in coastal waters.

The ROI strategy is a compelling and creative way to group in situ data and match the grouped data with other datasets for algorithm training. The strategy is explained well in this manuscript, and certainly has potential advantages over fixed spatiotemporal binning. OceanSODA-MDB is easily accessible as a series of NetCDF files. It provides co-located

satellite, model, reanalysis, and in situ data that can surely be used to re-train various algorithms that are in use today. This manuscript is a valuable contribution to the literature in its current form, but I've added a few comments and suggested corrections in the following sections.

Specific Comments

$CO_2$ system calculations only performed from $C_T$-$A_T$:

In lines 112–113 it is stated that carbonate chemistry calculations were only performed when $C_T$ and $A_T$ were available. Are there a significant number of cases for which other carbonate chemistry pairs were available (e.g., $C_T$-pH, $A_T$-$pCO_2$, etc.)? I'd be interested in the reasoning for only making carbonate chemistry calculations with the $C_T$-$A_T$ pair. Similar accuracy in calculated parameters should be obtainable by pairing either $C_T$ or $A_T$ with either pH or $pCO_2$ when the measurements are of sufficient quality (e.g., Orr et al., 2018).

Here we could have created calculated datasets from whatever measurements are available, which would vary from data point to data point and of which there could be up to six values. These could be combined into one prediction either with a hierarchy of combinations or by averaging all available combinations into a single value. However, the accuracy of the predictions depends on both the pair chosen and the predicted parameter, and more importantly, the combination of differently calculated values into a single ROI would make it difficult to know which values had the greatest weight in the calculation. In the end we decided that it was better to be clear about where the calculated values come from, so we restricted the choice to TA and DIC, which are usually measured together. If we continue to receive feedback that doing the calculation otherwise would be more useful, we will certainly consider it for future versions.

Creating the radial in situ data:

I think the explanation of how the data grouping is performed is great. It is clearly a complex process but breaking it down step-by-step in section 3.2 is very helpful. I am curious about the prospect of expanding this methodology to subsurface data and also how easily the methodology might be adapted to form finer resolution ROIs. I hope the authors might consider making their code publicly available at some point in the future to facilitate adaptions of their methodology to other datasets and/or spatiotemporal resolutions.

The current MDB, including as it does many surface datasets such as SOCAT and the satellite data, is deliberately restricted to the surface to ensure all datasets are comparable. The methodology could be extended to a third spatial dimension, perhaps with a scale that increases with depth (nobody needs 10m resolution at a depth of 6000m), but this is not the focus of this work. The potential for extension to finer resolution ROIs was built into the

software from the start, and the next version of the MDB will be at finer resolution. The code will be freely available on publication.

Updates to OceanSODA-MDB:

It is indicated in section 6 how the MDB could be updated. Are they any concrete plans to make updates to OceanSODA-MDB at regular intervals in the future?

Regular updates depend on future availability of funding, but the next version is funded and is already in preparation.

Minor Comments and Technical Corrections

All will be corrected unless commented otherwise

Line 130: The references in this line are repeated.

Line 135: Should this be dataset no. 4? Additionally, I'd be interested in knowing how the Argo data were acquired. A monthly snapshot should have an associated DOI (e.g., doi.org/10.17882/42182#95967 for Sep. 2022). If files were individually downloaded, were individual profile data files used, or interpolated Sprof files? I think this would be very helpful information for anyone who wants to replicate your methodology.

All individual profile data files containing pH were downloaded automatically from the IFREMER archive, and all data from depth <10m (generally only one value) averaged.

Line 141: "Dickson et al." should be outside the parentheses here.

Lines 157–161: Is it safe to say that this paragraph and the entries for these datasets in Table 1 should be eliminated since they are added to OceanSODA-MDB along with CODAP-NA?

Lines 275–277: Are discrete $pCO_2$ measurements treated any differently that underway $pCO_2$ measurements? I'd image that in many cases discrete $pCO_2$ data points would spatiotemporally match with those in the SOCAT database, but may be discarded based on the criteria noted here.

Good point! No, they're not treated any differently so some may be eliminated if they're not in SOCAT. In future versions we will check for this.

Lines 381–382: I don't think it's obvious why the advent of Bio-Argo floats would cause the mean number of pH measurements per ROI to decrease rather than increase. I think I can infer: the 10-day sampling cycle of the floats and the fact that only one pH measurement is obtained in the upper 10 meters means one individual ROI is generally created for each float profile? Regardless, since this result seems counterintuitive on first glance, I think it should be explained here.

Your inference is correct, we will amend the text.

Figure 16: Should the green line in the legend be T14 fit, rather than TS13?

References: Lauvset et al., 2018 is not in the reference list.

**RC3: 'Reply on RC2', Anonymous Referee #3, 05 Oct 2022**

The authors have presented a method by which data can be placed into spatio-temporal bins (which they call regions) and the data within each bin can be averaged and characterized.  They have used this method with several existing data sets to produce a hybrid data set that has had some spatial averaging applied.  Even though the manuscript is well written and tackles an important issue, I believe the manuscript should be returned to the authors because it is not well motivated and doesn't demonstrate the efficacy of their proposed approach.  The authors should have an opportunity to rewrite the manuscript to address several key issues.

The problem that the authors are attempting to address is a complicated one involving biases in relationships between predictor data and model/regression/algorithm output that can arise when the training data are not homogeneously spaced in space and time.  While the authors have well-described this (difficult to describe) problem, their solution to the problem has several flaws:

First is that it is complicated and computationally intensive.  Most studies that tackle this problem do so in a paragraph, whereas these authors have dedicated an entire paper to the issue.  This wouldn't be a problem if the solution were meaningfully better than most solutions, but I don't think the authors have demonstrated that yet (see below).

We present a methodology that could be used by others to generate their own datasets with different scales, input data etc. The MDB presented was our best attempt to provide data on a scale that we considered most useful for global studies. We consider that for global studies, the use of a consistent methodology across all key variables is itself a distinct advantage over simpler methods applied *ad hoc* to one pair of variables.

Second is that their solution doesn't seem distinct from the approach that studies take, which is to focus on the critical predictor/target variable with the least data density and either construct bins that are focused around retaining that information or avoid binning altogether and remap the more highly-resolved data onto the spatial-temporal locations of these data.  Note for example that the average number of AT measurements per bin is around 2 after WOCE suggesting that binning accomplished little for this variable.  Given the nominally-30 nm spacing (i.e., <100 km) for many open ocean cruises, this likely just means that adjacent stations along a cruise were averaged in most cases.  This approach will have done little or nothing to address the massive variability in data density regionally and seasonally, which can still be seen clearly

in Figures 8 through 14.  It is perhaps useful for pCO2 despite these limitations due to the extreme disparity in pCO2 data density on the spatial scales proposed by the authors... yet this also has not yet been demonstrated by the AT regression analysis.

AT and DIC, being entirely lab-based measurements, of course have low data density, and so the binning has little effect on them taken in isolation, but the method provides a consistent way to compare variables with highly divergent data densities (e.g. AT and pCO2w), and the optimisation of the ROI positions, though computationally intensive, reduces the probability of edge effects in fixed bin positions, e.g. an AT measurement at the edge of one bin not being compared to a cluster of pCO2w measurements at the near edge of an adjacent bin despite their close proximity. It is unfortunate that the application we chose does not use this comparison, but our aim was to find a simple global application (otherwise it would be a paper in its own right rather than an illustration) that might also be of use to some readers.

Third, having been created without a specific application in mind (or at least without one stated), it seems unlikely that this approach would be optimal for many studies.  Binning data always involves some loss of information, so the binning strategy must always be chosen to match the intended application.  I doubt that the decisions made in this study would be widely applicable and, when the approach is applicable, it might not be necessary for the reason noted above.  I was hoping the binning would have a way of sizing the bins according to the unique information content within a region, as perhaps represented by the heterogeneity of physical measurements within the bin (perhaps allowing smaller bins near fronts or coasts).

The ability to vary the region size is built into the software, making it as easy as possible to create custom datasets with different sizes, e. g. small, short duration regions for a regional study in a coastal or dynamic area. Any generic (rather than subjectively chosen) maximum region size adjustments would have to have a clear rationale that applies to all datasets included (not just in situ) and would make it more difficult to interpret the statistics. For this first attempt at a global MDB incorporating datasets of diverse origin and inherent scale, we consider that a fixed region size is appropriate.

Fourth, (and this is my strongest objection and the ones I would most hope the authors would address if they do have a chance for edits) the utility of the method has not been demonstrated. The authors provide an "apples to oranges" comparison refit of an alkalinity regression, but too many things are changed between the original publication and this analysis, making the comparison unhelpful.  Instead, the authors should compare fits of the T14 regressions made before and after binning the data in the manner that they suggest, ideally also comparing to regressions made using alternative binning strategies that would, hopefully, show the wisdom of using the spatio-temporal binning strategy that they employ herein.

The T14 reanalysis was intended as an illustrative example rather than a piece of work in its own right, so the comparison with the original work is intended as a guide to those who might consider using the new regressions rather than a claim of major progress. As you say below,

T14 is trained on the top 50m rather than the top 10m, so they would not be directly comparable even if they used the same AT data. However, a user with a focus on the top 10m (e. g. for satellite comparison work) might well be interested in the improvement over using T14 in this layer. We agree that AT is not ideal to illustrate the merits of the MDB over other binning methods, the low region occupancy meaning that any comparison will show minimal differences, but we struggled to find a sufficiently simple global example based on pCO2w, which has many more dependencies than AT, with the result that simple empirical pCO2w algorithms are generally regional rather than global.

I wonder if a schematic figure around line 300 could help the presentation of ROI creation methodology (I'm not sure what this would look like, but I found myself wishing for one here).

We have used several such figures in presentations of this work, but we couldn't find a way to usefully condense them into a single figure - even in several Powerpoint slides with animations, it's difficult to show the process in both space and time. However, it is a very visual process, so we agree that a visual aid would be useful - perhaps we can include something in Supplementary.

The authors should be commended for securing data from some rare data sources, but the fact that these original data are not included in the data availability section precludes this manuscript from publication in ESSD, if my understanding is correct.

One has now been published, the other is available on request from the author.

Line by line comments:

These will be corrected except where noted.

113: information required to constrain the carbonate system is missing from the list of provided constraints (e.g., nutrients and carbonate constant sets).

These were set to their SeaCarb defaults. In future versions we will use measured nutrients where available, otherwise climatological values.

120: how are these uncertainties expressed?  Standard uncertainties?  Confidence intervals?

Standard uncertainties

121: I'm curious about the number 0.005 for pH.  It seems quite low.  Was this meant to have been taken from Table 3 in the listed publication?  That table indicates 0.01.  Also, that table has the explicit caution that: "Note that these limits are not uncertainties but rather a priori estimates of global inter-cruise consistency…."

The reference is incorrect, the value was taken from Sulpis et al (2020), of which Siv Lauvset is a co-author, which itself references Johnson et al (2017). As a precaution, following Lauvset et al (2021), the uncertainty of pH will be set to 0.01 in future versions.

135: what uncertainties are assumed for these data?

An uncertainty estimate is included in each Bio-ARGO profile.

140: same question after noting that CRMs are meant to be checks on calibration and not themselves calibration materials

Instruments do need calibrating with known concentrations of DIC and TA, as stated in the user manual. At present the Dickson CRMs are the only quality assured standards available and hence are routinely used for calibration in the carbonate chemistry analysis community, hence it is written this way in nearly all publications. PML also run secondary standards to give an independent test of accuracy. The accuracies are reported in the text as better than $\pm2$ µmol/kg. In addition to the checks using CRMs to establish accuracy, drift, etc, PML also run duplicate and triplicate samples to test for precision. Again, precision is reported in the text as better than $\pm2$ µmol/kg.

145: were unpurified dyes used for these measurements, and, if so, what attempt is made to correct for pH biases from dye impurities? Alternatively, what uncertainty was assumed that accounts for these impurities?

Absorbance readings were corrected for the pH-change caused by the addition of the dye and pH is reported on the "Total" activity scale (pHT) (Dickson et al., 2007). The precision of triplicate pHT samples was ±0.001 units or better (triplicate samples were analysed on 17 occasions).

151: same question

None of the authors of this paper analyzed the data so we are limited to quoting values that are reported in the paper associated with the data.

159: how were these uncertainties assessed?

None of the authors of this paper analyzed the data so we are limited to quoting values that are reported in the paper associated with the data.

163: If these data are not publicly available then they cannot be used in an ESSD publication, correct?

The data have now been published, we will amend the text accordingly.

Polukhin, A. (2019). The role of river runoff in the Kara Sea surface layer acidification and carbonate system changes. *Environmental Research Letters*, *14*(10), 105007. DOI 10.1088/1748-9326/ab421e

165: what uncertainties are assumed for these methods?

AT accuracy is ~4 µM. The titre of the hydrochloric acid was analyzed using a standard soda solution (Na2CO3 of 99.995% purity) prepared by weighing, including the vacuum correction. Accuracy for pH (NBS scale) =is ~0.04 units (calibration with Standard-titres material dose for making 2nd class standard buffer solutions 4.01, 6.86, 9.18).

171: see earlier comment on accessibility

Data are available on request from Irina Pipko (irina@poi.dvo.ru).

181: what was the accuracy?

227: extra space before comma

266: how do you merge two datasets if they have different subdivisions?

They are merged to the finest scale needed, so monthly data will be divided into days before merging with daily data etc. We will clarify in the text.

286: There is now CO2SYS code that allows the direct propagation of uncertainties through the carbonate system calculations.  This code also has the advantage that it includes uncertainties in the carbonate constants which seem to be omitted from the present analysis aside from the random selection of constant sets.

At the time this work was formulated, we were not aware of the CO2SYS uncertainty propagation option (if it existed), so we attempted a similar procedure using the tools available in SeaCARB, which only allowed a choice between published sets. For the next version, which is in preparation, we are using CO2SYS uncertainty propagation.

304: RsOI

It is common practice to form a plural from an acronym by appending an 's' (e.g. MPs for Members of Parliament, not MsP), and we also consider it more readable.

310: recommended "expand the radius"

As well as radius, there is also temporal extension (think of it as a cylinder in space-time)

367: period used in AT number whereas comma used in other numbers

370: Bio ARGO is a program rather than a sensor

372: there is also a delay before the data are processed through annual GLODAP releases. You can find more recent cruise data at CCHDO.UCSD.EDU and other data repositories.

420: explain this more… why are these time series labeled as 0 and 1?

So that they are treated as separate datasets, in the same way as different years. Years are labelled with their year number, HOT and BATS are labelled 0 and 1, hence are treated as early years (low data density) in the grouping.

415: I had difficulty following the logic of this paragraph.  Mostly, it seemed to me that the work that had been done to create the combined data product is undermined by the complexity of this analysis, which seemed to be indicating that a lot of additional work was required to further divide the data product into subsets.

The further division is at the 'user' stage - having postulated that AT can be described regionally as a linear function of S+NO3 (T14), we use the MDB to note that, as well as the divisions in T14, there is also a difference in fit between open ocean and coastal, so we (as MDB users) divide the algorithm to form separate regressions. Again as MDB users, we further divide each region's data into subsets to perform cross validation of our algorithm.This has nothing to do with the data division used to form the MDB in the first place, and different users will divide the MDB data in different ways depending on their application, but in all cases the initial MDB division serves to minimise the impact of differences in data density on the statistics of such analyses.

430: how large are the validation subsets on average?  Have you done anything to avoid including measurements from a single cruise in both the training and the validation data?  This is usually considered an important practice because of the strong temporal correlations between measurements on a single cruise.

We did this by dividing the data by year, which does leave open the possibility that a cruise spanning the end of the year could be included in both training and validation datasets, however this should happen rarely enough to have a minimal effect on the statistics. The next version includes EXPOCODE in the MDB, which will allow tighter control of this.

435: I also had some difficulty understanding what was being done in this paragraph, or why exactly.  Why are the authors so motivated to include nearshore data that it would justify this extra text and complexity.  This is just a proof-of-concept demonstration, so does it need to be comprehensive in terms of including coastal data?  Have the authors shown the consequences of simply including all coastal data?

In order to provide something as useful as possible from this example application, our aim was to provide an algorithm with global scope, and having identified from the MDB that coastal data sometimes (but not always) follows a different trend, this was an attempt to use the whole MDB to minimise the global RMSE by creating coastal subdivisions in each T14 region where needed, rather than exclude (or include at the expense of increasing open ocean uncertainty) coastal data. The methodology ended up quite complicated, but the result is simple - in each

T14 region there's a distance and/or depth threshold beyond which a different regression is used.

447: this is not a straight comparison since T14 use data down to 50 m in their training product the algorithms are therefore producing different estimates (and from different predictor data as well, I think, but I'm not sure). A true proof of concept would require creating another version of the algorithm using the same training and validation data set, but using the binning approach of Takahashi et al. 2014

This would be appropriate if updating T14 were the main aim of the paper but is not necessary for an example application. The predictor data are indeed different, since the MDB uses data up to 2020. The example application could be thought of as 'how can we use the MDB to update T14 for surface (<10m) waters'.

Figure 2: should we be concerned by the large number of TA measurements per bin in ~1978? Why are there so many more measurements per bin in the earlier portions?

Well spotted, there's an error in this plot (and probably others of this type). We will update the plots.

CC1: 'Citation of R packages', Jean-Pierre Gattuso, 12 Aug 2022

While I am reading your interesting contribution, I note that you have used an R package without a proper citation. It is important to give credit to authors of packages who make their work openly accessible to the community but few people know how to do it.

Thank you for your comment and explanation, we will cite SeaCarb as suggested.